# Broadly conserved FlgV controls flagellar assembly and *Borrelia burgdorferi* dissemination in mice

Maxime Zamba-Campero[1,8], Daniel Soliman [1,8], Huaxin Yu[2,3], Amanda G. Lasseter[4], Yuen-Yan Chang [1], Julia L. Silberman [1], Jun Liu [2,3], L. Aravind[5], Mollie W. Jewett [4], Gisela Storz [1] & Philip P. Adams [1,4,6,7] ✉

Flagella propel pathogens through their environments, yet are expensive to synthesize and are immunogenic. Thus, complex hierarchical regulatory networks control flagellar gene expression. Spirochetes are highly motile bacteria, but peculiarly, the archetypal flagellar regulator $\sigma^{28}$ is absent in the Lyme spirochete *Borrelia burgdorferi*. Here, we show that gene *bb0268* (*flgV*) in *B. burgdorferi*, previously and incorrectly annotated to encode the RNA-binding protein Hfq, is instead a structural flagellar component that modulates flagellar assembly. The *flgV* gene is broadly conserved in the flagellar super-operon alongside $\sigma^{28}$ in many *Spirochaetae*, *Firmicutes* and other phyla, with distant homologs in *Epsilonproteobacteria*. We find that *B. burgdorferi* FlgV is localized within flagellar basal bodies, and strains lacking *flgV* produce fewer and shorter flagellar filaments and are defective in cell division and motility. During the enzootic cycle, *flgV*-deficient *B. burgdorferi* survive and replicate in *Ixodes* ticks but are attenuated for infection and dissemination in mice. Our work defines infection timepoints when spirochete motility is most crucial and implicates FlgV as a broadly distributed structural flagellar component that modulates flagellar assembly.

Flagella are complex membrane-embedded nanomachines with a protruding filament that allow bacteria to move through liquids and across solid surfaces. Flagella allow a broad range of microorganisms to inhabit different environmental niches, to migrate toward nutrient-rich regions, to escape unfavorable conditions, and to move from host to host. For several bacteria, flagella are even connected to basic processes such as cell division and maintenance of cell shape (reviewed in refs. 1–3). Unique endoflagella, which are anchored at each cell pole and extend through the periplasmic space, rather than

the extracellular environment, are hallmarks of Spirochaetota (reviewed in refs. 4,5). It is this flagellar placement that creates the flat-wave structure of the spirochetal bacterium *Borrelia* (*Borreliella*) *burgdorferi*[6], the Lyme disease pathogen.

Lyme disease is an emerging infectious disease and the foremost vector-borne illness in the United States, with almost half a million infections estimated annually[7]. *B. burgdorferi* exists in a complex enzootic cycle that requires acquisition and transmission of the spirochete between *Ixodes scapularis* ticks and small mammals and birds

[1]Division of Molecular and Cellular Biology, Eunice Kennedy Shriver National Institute of Child Health and Human Development, National Institutes of Health, Bethesda, MD 20892, USA. [2]Department of Microbial Pathogenesis, Yale School of Medicine, New Haven, CT 06536, USA. [3]Microbial Sciences Institute, Yale University, West Haven, CT 06516, USA. [4]Division of Immunity and Pathogenesis, Burnett School of Biomedical Sciences, University of Central Florida College of Medicine, Orlando, FL 32827, USA. [5]Division of Intramural Research, National Library of Medicine, National Institutes of Health, Bethesda, MD 20894, USA. [6]Postdoctoral Research Associate Program, National Institute of General Medical Sciences, National Institutes of Health, Bethesda, MD 20892, USA. [7]Independent Research Scholar Program, Intramural Research Program, National Institutes of Health, Bethesda, MD 20892, USA. [8]These authors contributed equally: Maxime Zamba-Campero, Daniel Soliman. ✉e-mail: philip.adams@nih.gov

(reviewed in ref. 8). Motility of *B. burgdorferi* is essential for survival throughout the enzootic cycle[9,10]. Larval ticks hatch uninfected and acquire *B. burgdorferi* upon feeding on an infected host, most commonly the white-footed mouse in the northeastern United States. During larval-tick feeding, the bacterium navigates from the mouse dermis to colonize the tick midgut. The tick will then molt into a nymph, infected with *B. burgdorferi*. Inside the unfed nymph, spirochetes remain sessile to survive the nutrient-poor midgut and await the next bloodmeal, which can occur months to a year after the molt. When the nymphal tick bites its next host, spirochetes associate with the basement membrane of the midgut as nonmotile aggregates and then become motile, migrating from the tick midgut to the tick salivary glands[11], ultimately infecting the skin of a vertebrate host. In the mammal, *B. burgdorferi* spread from the initial bite site through the blood (hematogenous dissemination) and other routes, to colonize distal tissues. Overall, to progress through this enzootic cycle, *B. burgdorferi* must finely tune its motility to conserve energy when needed and disseminate when able.

Regulation of bacterial motility, which can be at the level of expression of flagellar proteins or the functional level of flagellar motor rotation and chemotaxis, has been studied most in Enterobacteriaceae and Bacillaceae, (reviewed in refs. 12–14). In these model organisms, the genes encoding the basal body, rod and hook of the nascent flagellum and the alternative sigma factor, $\sigma^{28}$ ($\sigma^F$, FliA, SigD), are transcribed first. Once the hook–basal body is assembled, RNA polymerase with $\sigma^{28}$ transcribes the genes encoding the flagellar filament. This hierarchical gene regulation prevents malformed hook–basal body complexes and allows a bacterium to rapidly control motility in an energetically efficient way. The genus *Borrelia/Borelliella*, but not other spirochetes (e.g., *Treponema*, *Spirochaeta*, *Leptospira*, and *Turneriella*) lacks the gene encoding $\sigma^{28}$. Thus, all *B. burgdorferi* flagellar genes were reported to be transcribed constitutively under the control of the housekeeping sigma factor, $\sigma^{70}$; the first bacterium discovered to have this scheme of flagellar gene expression[15,16].

The conserved and generally-recognized motility "superoperon" is denoted the "*flgB* superoperon" in *B. burgdorferi* and includes 31 genes (*bb0294* to *bb0264*)[16,17]. These genes encode the components of the flagellar structure (basal body, hook, rod and filaments), motor ATPases, and flagellar assembly proteins. Cryo-electron tomography has visualized the in situ flagellar basal body structure and sequential assembly of *B. burgdorferi* periplasmic flagella[18], yet the regulation of this temporal assembly and how environmental signals impact flagellar biosynthesis remain poorly understood.

In this study we characterize gene *bb0268* in the "*flgB* superoperon". BB0268 was previously annotated as a homolog of Hfq[19]. Hfq, which is best characterized in Enterobacteriaceae, is a homohexameric RNA binding protein of the Sm domain superfamily that stabilizes small RNAs (sRNAs) and facilitates sRNA base pairing with cognate RNA targets in many bacteria (reviewed in refs. 20,21). In our recharacterization of BB0268, we demonstrated that the protein is evolutionarily unrelated to Hfq. BB0268 has an entirely different structure from Hfq and does not bind RNA. Comparative genomics analysis of Hfq suggests *B. burgdorferi* lacks an Hfq homolog entirely. Instead, we found strong co-conservation of BB0268 homologs with the genes encoding the flagellar regulators FlhF, FlhG, and $\sigma^{28}$, spanning the genomes of numerous flagellated bacteria, and thus we renamed the *bb0268* gene as *flgV*. We discovered FlgV localizes within the *B. burgdorferi* flagellar basal body and impacts the number and length of flagellar filaments, as well as *B. burgdorferi* dissemination during mouse infection. We therefore propose that FlgV plays previously uncharacterized structural and functional roles in late-stage flagellar assembly.

## Results

### BB0268 was misannotated as an RNA-binding protein

*B. burgdorferi* harbors a large 26.3 kb "*flgB* superoperon" which encompasses 31 flagellar biosynthesis genes, including *flhF* (*bb0270*) and *flhG* (*bb0269*), potential regulators of flagellar biosynthesis[17]. One study previously reported that *bb0268* (present directly downstream of *flhF* and *flhG*) encodes an "atypical Hfq homolog"[19]. Hfq, an ortholog of the Sm proteins found in archaea and eukaryotes[22], is the most-extensively studied bacterial regulatory RNA-binding protein. In many bacteria, the Hfq hexamer uses distinct surfaces to bind specific sequence motifs in small RNAs (sRNAs) and their target RNAs, often resulting in regulatory consequences (reviewed in refs. 20,21).

Given the report that BB0268 is an Hfq homolog, we sought to identify possible RNAs associated with BB0268. We assayed the ability of BB0268 to bind RNA in vivo by performing a co-immunoprecipitation (co-IP) experiment with *B. burgdorferi* expressing a C-terminal 3XFLAG tagged derivative of BB0268, encoded at the endogenous chromosomal location. The parent wild-type (WT) strain was used as a control. No growth defect was observed in the BB0268-3XFLAG strain compared to WT (Fig. S1A). Additionally, in the same experiment, we used *E. coli* cells expressing a single-FLAG-tagged version of the *E. coli* Hfq[23], also encoded at the endogenous locus, and the corresponding *E. coli* WT strain. The tagged proteins were immunoprecipitated (IP) and RNA was isolated from the IP samples. Immunoblot analysis of cell lysate (total), supernatant, and elution (IP) samples confirmed the production of both BB0268-3XFLAG and *E. coli* Hfq-FLAG, which were enriched by IP (Fig. 1A). The RNA isolated from the IP samples was analyzed on an Agilent TapeStation system (Fig. 1B). Insignificant RNA levels were detected with BB0268-3XFLAG IP compared to ~17 ng/µl IP RNA with *E. coli* Hfq-FLAG.

In a separate experiment, we used BB0268 antibodies (generated for this study; see Methods) to immunoprecipitate native BB0268. Again, we observed enrichment of BB0268 in the IP fraction by immunoblot analysis (Fig. S1B), but no RNA was detected (Fig. 1C).

To demonstrate that this co-IP approach can successfully enrich for RNA associated with a bona fide RNA-binding protein in *B. burgdorferi* we assayed the KH-domain protein, KhpB. KH domain proteins are another class of RNA-binding proteins in bacteria, and *B. burgdorferi* harbors KhpA (BB0696) and KhpB (BB0443) homologs (reviewed in ref. 24). We observed enrichment of KhpB in the IP fraction by immunoblot analysis with antibodies targeting native KhpB (Fig. S1C). In contrast to BB0268, we detected ~8 ng/µl IP RNA with KhpB (Fig. 1D).

It was previously reported[19] that recombinant BB0268 binds in vitro transcribed *B. burgdorferi* SR0440, a possible *B. burgdorferi* base-pairing sRNA denoted DsrA[25] based on the well-characterized DsrA sRNA in *E. coli*[26]. However, the electrophoretic mobility shift assays did not include negative controls and used high amounts of purified BB0268 (a maximum protein:RNA ratio of ~823:1 molecules[19], compared to the maximum ~15:1 ratio typically used for *E. coli* Hfq experiments[22]). To test if BB0268 specifically interacts with SR0440, we carried out northern analysis on the *B. burgdorferi* BB0268-3XFLAG total, supernatant, and IP samples. In our experiment, no SR0440 was detected in the BB0268-3XFLAG IP sample (Fig. S1D). In contrast, we observed a clear enrichment of *E. coli* DsrA in the *E. coli* Hfq-FLAG IP sample (Fig. S1E). Collectively, these data indicate that BB0268 does not globally bind RNA or specifically bind a *B. burgdorferi* sRNA.

### BB0268 is not an Hfq homolog

A recent review conducted a systematic phyletic survey of Hfq homologs across 628 bacterial species representing all major phyla[27]. This identified an Hfq in two Leptospiraceae, *Leptospira* and *Turneriella*, but not in Borreliae. To ensure that no divergent version had been missed, we carried out a systematic search for Hfq homologs using sensitive sequence profile searches. These detected not only diverse bacterial Hfq proteins but also significantly identified the divergent archaeo-eukaryotic Sm clade proteins. Phylogenetic and divergence rate analysis of the prokaryotic Hfq proteins revealed that they fell into two broad clades: (i) the classic Hfq (Fig. S2A); and (ii) the

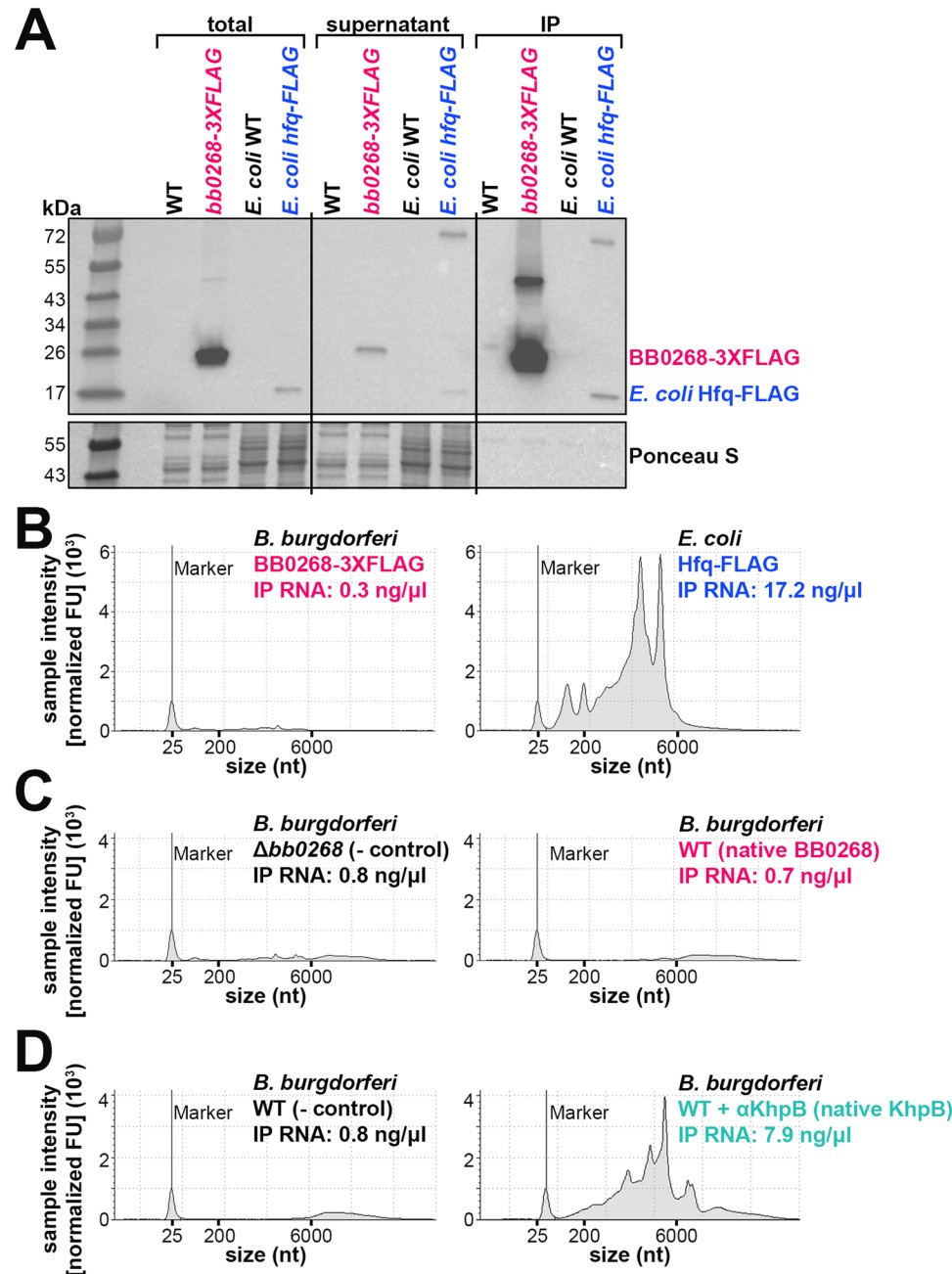

**Fig. 1 | BB0268 does not bind RNA. A** Immunoblot analysis of immunoprecipitated *B. burgdorferi* BB0268-3XFLAG (PA007) or *E. coli* Hfq-FLAG (EC153) samples. The parent WT strains (PA003 and EC152, respectively) were used as controls. *B. burgdorferi* cells were grown to $1 \times 10^8$ cells/ml and *E. coli* cells were grown to an $OD_{600} = 1.0$ and then exposed to UV to crosslink any RNAs associated with the proteins. After cell lysis, the tagged proteins were immunoprecipitated (IP) and RNA was isolated from the IP samples. Equal volumes of cell lysis (total), supernatant of the α-FLAG-beads during washing, and elution from the α-FLAG-beads (IP) were separated on a Tris-Glycine gel, transferred to a membrane, stained with Ponceau S, and probed using α-FLAG antibodies. Size markers are indicated. **B** Quantification of α-FLAG immunoprecipitated RNA. Each sample (2 μl) was analyzed using an Agilent 4200 TapeStation System and the RNA high sensitivity reagents. **C** Quantification of α-BB0268 immunoprecipitated RNA, as in panel (**B**). **D** Quantification of α-KhpB immunoprecipitated RNA, as in panel (**B**). Panels **A** and **B** are a representative experiment from three independent experimental replicates, also including lysates from stationary phase *B. burgdorferi*. All replicates showed the same results.

mobile Hfq (Fig. S2B) clades. The classic Hfq clade is typified by the *E. coli* and *B. subtilis* Hfq proteins and is found in several major bacterial lineages, including Spirochaeta, such as Leptospiraceae. In the classic clade, *hfq* has gene-neighbors that are conserved across distant phyla including *miaA* (a tRNA modification gene), *hflX/hflK/hflC* (genes involved in ribosome release and as protein stability factors functioning with FtsH) and *mutL/mutS* (genes involved in DNA repair/ribosome rescue and release). The mobile Hfq clade is the only type of

Hfq found in cyanobacteria and has been widely disseminated through lateral gene transfer across bacteria from an archaeal progenitor, probably through the medium of bacteriophages and plasmids, some of which also encode the protein. Sequence profiles for both these clades generated no statistically significant hits from the *B. burgdorferi* proteome establishing beyond a doubt that Hfq is absent in this organism. Further, the locus encoding *miaA* (*bbO821*; Fig. S2C), showed no gene in the position usually occupied by Hfq homologs.

The lack of RNA-binding activity for BB0268 and our conservation analysis were consistent that Hfq is missing from *B. burgdorferi*, therefore, we looked further into the original *bb0268* study which reported BB0268 as an Hfq homolog[19]. The study predicted that *B. burgdorferi* BB0268 has a protein structure similar to *E. coli* and *S. aureus* Hfq. However, this is incompatible with AlphaFold 3[28] structure predictions for monomers of *E. coli* Hfq (Fig. S3A) and BB0268 (Fig. S3B) which showed no similarity between the two proteins. Hfq harbors an SH3-like β-sheet fold structure and forms a hexamer (Fig. S3C), reviewed in ref. 29. In contrast, we predict that BB0268 harbors two transmembrane helices followed by an unstructured cytoplasmic tail and forms a dimer (Fig. S3D), discussed below.

It was also previously reported that *bb0268* complements an *E. coli* Δ*hfq* strain[19]. Thus, we examined the effects of producing BB0268 in *E. coli*. We performed a growth curve using *E. coli* Δ*hfq* cells, but Δ*hfq* cells producing BB0268 grew worse than WT and *bb0268* did not complement the *hfq*-dependent growth defect (Fig. S4A). The levels of RpoS, the general stress-responsive σ factor (reviewed in ref. 30) are known to decrease in an *E. coli* Δ*hfq* strain[31]. It was previously proposed that *bb0268* complementation restored levels of RpoS, as measured by β-galactosidase activity from a *rpoS-lacZ* fusion in an *E. coli* Δ*hfq* background[19]. To repeat this experiment, we examined *E. coli* RpoS levels using antibodies to the native protein (Fig. S4B). RpoS decreased in Δ*hfq* samples at logarithmic and stationary phase, as expected, but we did not observe restoration of *E. coli* RpoS levels upon expression of *bb0268*. BB0268 production did lead to a modest RpoS increase in the stationary phase samples; however, it is unknown if this effect is direct or caused from stress responses invoked by BB0268 synthesis in *E. coli*.

The major role of *E. coli* Hfq is to facilitate RNA–RNA interactions and stabilize sRNAs, preventing them from degradation by RNases (reviewed in refs. 20,21), but complementation of these *E. coli* Δ*hfq* phenotypes by *bb0268* was never tested. We isolated RNA from the same cultures used for immunoblot analysis (Fig. S4B) and performed a northern analysis to measure the levels of various sRNAs (Fig. S4C). We probed for the sRNA ChiX, one of the most enriched sRNAs bound by *E. coli* Hfq during growth in LB and highly unstable in an Δ*hfq* background[23]. No complementation was observed for ChiX levels in the Δ*hfq* cells expressing *bb0268*. We also examined the levels of two RpoS-dependent sRNAs, DsrA and SdsR[26,32]. Expression of *bb0268* in the Δ*hfq* strain failed to complement the effect of Δ*hfq* on sRNA levels in the logarithmic phase samples. At stationary phase, DsrA and SdsR bands were detected, albeit at negligible levels compared to the WT and the *hfq* complementation samples at the same time point. Collectively, these data demonstrated that ectopic expression of *bb0268* in *E. coli* does not significantly complement the effects of an *hfq* deletion mutant and the function of BB0268 is disparate from that of Hfq.

## *bb0268* is co-conserved with flagellar genes and σ[28] across bacteria

To understand the true function of BB0268, we examined RNA expression and the genomic context of *bb0268*. The gene is embedded within the "*flgB* superoperon" and expression of the entire operon has been suggested to be constitutive, driven by a consensus σ[70] promoter upstream of *flgB*[16,17]. However, *B. burgdorferi* transcriptome mapping studies[33,34] documented three predominant transcription start sites (TSSs), several 5′ processed ends, and several 3′ ends throughout the operon (Fig. S5A). Therefore, this genomic region likely encodes multiple RNA products.

Northern analysis with a probe internal to *bb0268* (pink asterisk; Fig. S5A) indicated that the gene is expressed in multiple transcripts across growth ranging in length from ~500 to 6,000 nt (Fig. S5B). No *bb0268* transcripts were detected by northern analysis using RNA isolated from a *B. burgdorferi* Δ*bb0268* strain in which the entire *bb0268* ORF (480 bp) was replaced with a streptomycin (*aadA*)

resistance cassette (1159 bp). Additional northern analysis with probes internal to the *bb0268*-neighboring genes *fapA* (*bb0267*), *flhG* (*bb0269*), and *flhB* (*bb0272*) (black asterisks; S5A and S5B) also detected multiple transcripts including a ~ 6,000 nt band. Some transcripts detected with the *fapA* (*bb0267*) probe increased in the *bb0268* deletion samples, likely from the presence of the promoter for the *aadA* resistance cassette, which is oriented in the same direction of the operon. A probe internal to the *aadA* ORF suggested the *aadA* resistance cassette transcript can be terminated, cleaved, or co-expressed with the surrounding genes in the operon (Fig. S5B). However, immunoblot analysis using antibodies to native FapA (Fig. S5C) showed FapA levels in WT and Δ*bb0268* lysates are equivalent (Fig. S5D). Given the complexity of this operon, it is challenging to define the exact transcriptional units/RNAs, nevertheless, these data document that the operon is expressed as multi- and single-gene products, with *bb0268* co-transcribed with *bb0272*–*bb0264*, and that the Δ*bb0268* mutation does not have polar effects.

We next examined the conservation of BB0268 across bacteria by performing iterative and transitive sequence-profile and Hidden Markov Model searches (see Methods). These searches recovered a widespread family of BB0268 homologs with significant e-values ($e < 10^{-5}$) from spirochetes, the PVC (Planctomyces, Verrucomicrobia, Chlamydia, Phycisphaerae, Omnitrophota) clade, flagellated members of the firmicute phylum (Bacillota), Nitrospinota, a subset of motile Chloroflexi, and certain other poorly studied bacterial clades (Desantisbacteria, Delongbacteria, Margulisbacteria, Glassbacteria, Ozemobacteria, Poribacteria). An analysis of the multiple sequence alignment and the predicted structure of these proteins (Fig. S6A) revealed a two transmembrane (2TM) architecture with a well-conserved intramembrane arginine residue in the second transmembrane helix followed by a disordered, highly polar cytoplasmic tail of variable length culminating in a short, conserved C-terminal peptide with multiple hydrophobic amino acids and a basic residue.

Despite its broad phyletic distribution, the *bb0268* family has not been well characterized. Hence, we conducted a systematic analysis of the gene neighborhoods of this family to infer potential functional connections (Fig. 2; predicted protein domains labeled for each gene). Without exceptions, all members of this superfamily are located within the flagellar superoperon across their broad phyletic distribution. Within this superoperon, the *bb0268* gene occurs at the junction between two genetic sub-modules for flagellar biosynthesis. First, there is a persistent association with the sub-module encoding two flagellar assembly proteins, FlhF (a flagellar GTPase), which regulates flagella number and structural organization[17], and FlhG (FleN/MotR; a flagellar MinD NTPase homolog), which similarly has been implicated in the control of flagella count. Second, the *bb0268* gene associates with a sub-module encoding the flagellar σ factor, σ[28] (Fl-Sigma; lost in *Borrelia*), FapA (flagellar assembly protein A), YjfB (an uncharacterized small protein), and a flagellar-associated transmembrane protein with a helix-turn-helix domain (TM + HTH). These observations strongly implicated *bb0268* in flagellar function.

Interestingly, we also recovered homologs of a flagellar protein previously identified in the Campylobacterales (Campylobacter-Helicobacter), named FlgV, for its association with a motility phenotype in a *Campylobacter jejuni* transposon mutagenesis screen[35]. Structural analysis of the Campylobacterales FlgV proteins revealed a 2TM architecture with a C-terminal cytoplasmic disordered tail congruent to the BB0268 family, though at lower significance (e-value = 0.1). Importantly, genome-context analysis revealed that Campylobacterales *flgV* genes too were encoded in the flagellar operons associated with the same two flagellar genetic sub-modules as the *bona fide bb0268* homologs (Fig. 2). Taken together, these observations indicated the presence of a structurally similar 2TM protein in numerous bacteria. Phylogenetic analysis using the maximum likelihood method showed that these 2TM proteins belong to three distinct well-

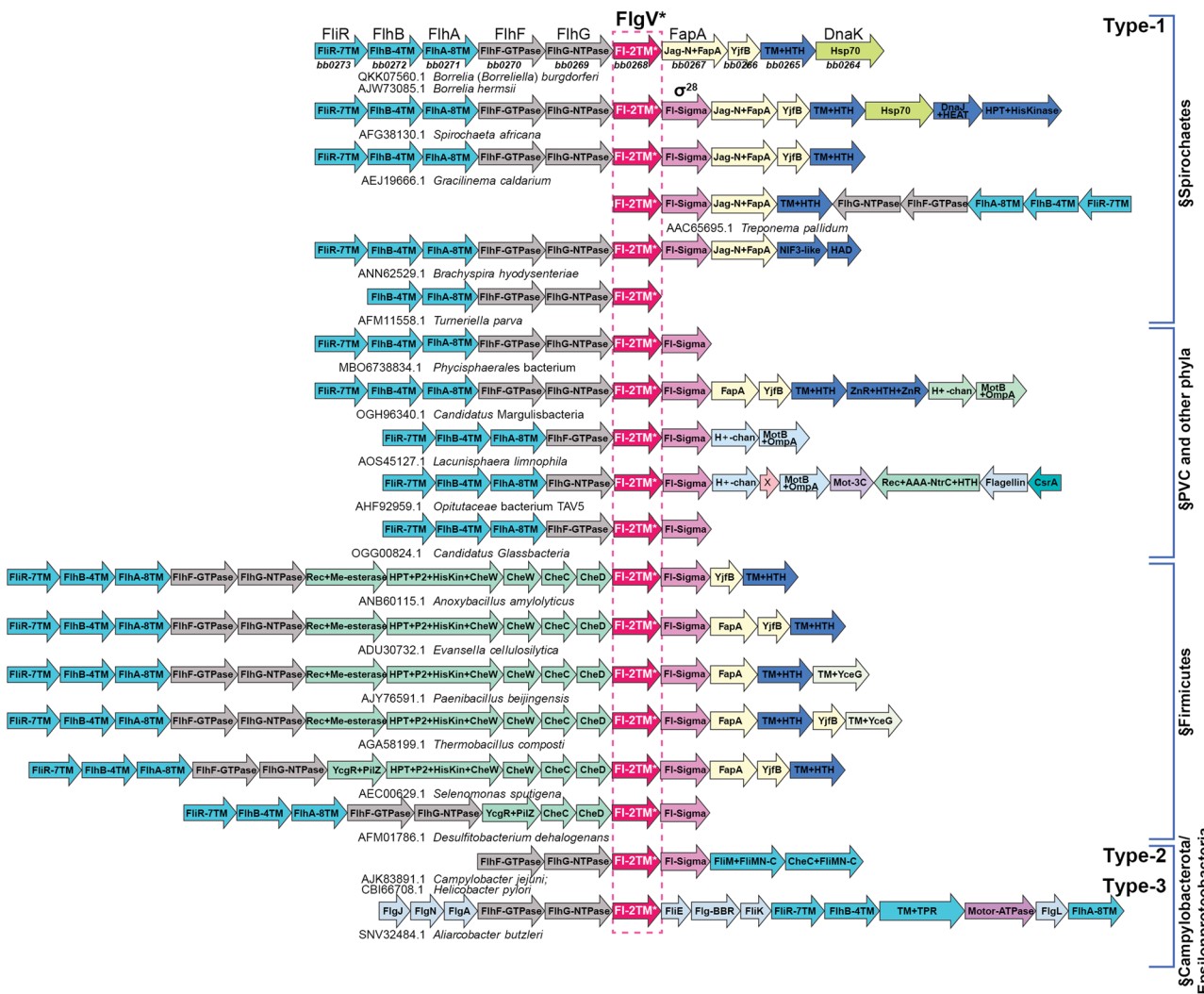

**Fig. 2 | *bb0268* (*flgV*) is broadly conserved with flagellar genes and σ²⁸ in bacteria.** Examples of the flagellar superoperons from diverse bacterial lineages mentioned in the text are depicted, centered on the gene encoding FlgV, marked with a dashed box. The genes are shown as box arrows (only drawn to approximate size) and labeled with the domain architectures of the encoded proteins. The genes belonging to distinct genetic/functional submodules are distinguished using different colors. Each operon is labeled, below the box arrows, based on the GenBank accession of the anchor *flgV* gene and the species name. The encoded protein names from model systems are shown above the first row, and the corresponding *Borrelia* (*Borreliella*) gene names are listed below the operon from that organism. The versions are labeled as Type-1, -2, and -3 based on the type of FlgV encoded in the operon. All versions are Type-1 except the last two exemplars. The *Turneriella parva* superoperon is fragmented into more than one gene cluster, and the one containing *flgV* ends with that gene.

supported clades (Fig. 2 and Fig. S7): (i) Type-1, which is the most widely distributed, is defined by the above-mentioned BBO268-related proteins and are encoded in flagellar superoperons primarily from spirochetes and Bacillota; (ii) Type-2, which includes the FlgV of Campylobacteria and Helicobacteria, typically are encoded in a short operon with just six flagellar genes; (iii) Type-3, which occur in Arcobacteria-like Campylobacterota and are encoded in the flagellar superoperon, but with a gene arrangement that is distinct from the first two types. Given its gene-neighborhood and protein structural equivalence, we renamed *B. burgdorferi bb0268* as *flgV*, keeping with the *Campylobacter jejuni* nomenclature. Despite being a widely distributed gene with a predicted flagellar function, *flgV* is absent in well-studied models such as *B. subtilis* and *E. coli*.

The *flgV* gene is frequently adjacent to the gene encoding σ²⁸, the critical regulator of flagellar filament synthesis across bacteria. While all Spirochaeta genomes contain *flgV*, it is striking that *Borrelia* are unique in that they specifically lack σ²⁸ (Fig. 2; *Turneriella* encodes σ²⁸ at a different location from *flgV*). Thus, how *Borrelia* sp. regulate late-stage assembly of flagella has remained elusive. Here we sought to test the hypothesis that *flgV* impacts motility and flagellar assembly in *B. burgdorferi*.

## FlgV levels impact cell division and motility

To characterize *B. burgdorferi flgV*, we analyzed cells lacking *flgV* and overexpressing the FlgV protein. Given the complicated transcriptional architecture of the *flgV* region, we complemented *flgV* in the deletion mutant by altering the standard pBSV2G *B. burgdorferi* shuttle vector[36] to have the IPTG-inducible lactose promoter and the *lacI* gene codon optimized for *B. burgdorferi*[37] (p_ind). This resulted in a set of isogenic strains: WT (WT/p_ind), *flgV* deletion (Δ*flgV*/p_ind), and *flgV* complement (Δ*flgV*/p_ind+*flgV*). To generate a *flgV* overexpression strain (++*flgV*) we cloned the *flgV* ORF sequence downstream of the constitutively active *flaB* promoter, into the standard pBSV2G vector in WT *B. burgdorferi* (WT/p_con ++*flgV*). WT spirochetes harboring the pBSV2G empty vector (WT/p) were used as a control for the ++*flgV* strain. We analyzed FlgV levels in these strains by immunoblot analysis, using antibodies to the native protein. FlgV was absent from the deletion strain and detected in the Δ*flgV*/p_ind+*flgV* strain, albeit at

slightly higher levels compared to WT (Fig. 3A). Significantly higher FlgV levels were observed with the ++*flgV* construct (Fig. 3B).

We first measured the growth of the *flgV* strains by dark-field microscopy enumeration (Fig. 3C). When *flgV* was deleted, the spirochetes had a moderate *flgV*-dependent lag in cell growth. A second growth curve with data collected approximately every 12 h demonstrated a significant, prolonged doubling time for Δ*flgV* spirochetes in lag phase (8.5 h), compared to WT spirochetes in lag phase (4.6 h) (Fig. S4F). Complementation by IPTG-induction of *flgV* resulted in full restoration of growth compared to the Δ*flgV*/p$_{ind}$ strain (Fig. 3C). There was no difference in growth between the two WT control strains.

We observed differences in cell morphology when *flgV* levels were altered. Spirochetes grown to exponential phase and lacking *flgV* appeared longer than WT *B. burgdorferi* (Fig. 3D, E), as reported previously[19]. Closer inspection of the longer Δ*flgV*/p$_{ind}$ cells showed conjoined Δ*flgV* spirochetes and the presence of septa (white arrows; Fig. 3E). The *flgV*-dependent morphology phenotypes were complemented with p$_{ind}$+*flgV* (Fig. 3F). Overproduction of FlgV also resulted in longer spirochetes, compared to WT cells (Fig. 3G, H). To quantify these observations, we measured the length of ~100 cells per strain using dark-field microscopy (Fig. 3I). Both the lack of and elevated levels of FlgV significantly increased the average spirochete length in exponential phase. There was no difference in the cell length of Δ*flgV*/p$_{ind}$ or WT/p$_{con}$ ++*flgV* *B. burgdorferi* in stationary phase, compared to the WT controls (Fig. S4G).

Cell division and motility are likely inherently linked in spirochetes, as efficient separation at the mid-cell septum requires functional flagella[38]. To measure the contribution of *flgV* to spirochete motility, we inoculated the *flgV* strains into semisolid plates (Fig. 3J) and measured the diameter of motility rings (Fig. 3K). On average, Δ*flgV*/p$_{ind}$ spirochetes formed smaller rings compared to WT spirochetes. The deletion phenotype was rescued by *flgV* complementation. A similar defect in motility was observed with spirochetes overproducing FlgV. Collectively our data indicate levels of FlgV must be tightly regulated, as both too much or too little protein resulted in cell division and motility defects.

## FlgV is membrane associated and localized to the cell poles and mid-cell

Members of the FlgV protein family (Fig. S6A) have a similar structure of two predicted N-terminal transmembrane helices (amino acids 13 to 35 and 40 to 57, in *B. burgdorferi*). To experimentally determine FlgV localization, total cell lysates from WT *B. burgdorferi* were fractionated into soluble and membrane portions and analyzed by immunoblot analysis (Fig. 4A). FlgV was restricted to the membrane fraction like membrane-associated outer surface protein C (OspC) and not like soluble superoxide dismutase A (SodA). Previous gel filtration chromatography reported that FlgV likely forms a dimer in vitro[19], consistent with this, we observed a higher molecular weight FlgV band (Fig. 4A) that was more apparent in our co-IP (Fig. 1A) and overexpression (Fig. 3B) analysis.

We created a FlgV C-terminal fusion to *B. burgdorferi* codon-optimized green fluorescent protein[39] at the native *flgV* locus (Fig. S6B). Spirochetes producing FlgV-GFP were grown to logarithmic phase and imaged by fluorescence confocal microscopy (Fig. 4B). We constructed a demograph where the GFP profile for 300 spirochetes was displayed for each cell, by aligning spirochetes in order of increasing length. FlgV-GFP localized to the cell poles, which overlaps with the location of the flagellar basal bodies[40]. The analysis also revealed GFP intensity at two puncta in the middle of each spirochete. The mid-cell coincides with the point of septation of a dividing spirochete, thus the new poles of a future daughter cell[41] and location of future flagella. In the longest spirochetes we examined, which may actually be two spirochetes about to divide, we sometimes observed two additional internal doublet-puncta of GFP intensity (Fig. 4B;

demograph), which could correspond to the future mid-cell of the two new spirochetes. Similar FlgV-GFP localization was observed for cells grown to stationary phase (Fig. S6C). Together, our observations support a hypothesis that membrane-associated FlgV co-localizes with functional flagellar basal bodies at the cell ends and nascent flagella basal bodies at the mid-cell.

## FlgV is a component of the flagellar basal body

To investigate a possible FlgV association with the flagellar basal body, we analyzed in situ structures of flagellar basal bodies using cryo-electron tomography (cryo-ET). The in situ structure from WT/p cells revealed major components of the flagellar basal body including the export apparatus, rotor complex (C and MS rings), stator complex, and rod (Fig. 4C, D), as previously defined[40]. We also determined in situ structures of the flagellar basal bodies from Δ*flgV*/p$_{ind}$ (Fig. 4E) and Δ*flgV*/p$_{ind}$+*flgV* (Fig. 4F) *B. burgdorferi*. The structures of the Δ*flgV* basal bodies exhibited many of the same features as the WT basal bodies; however, Δ*flgV* basal bodies lacked a density adjacent to the C and MS rings of the rotor (pink arrows; Fig. 4D, E). Complementation restored the absent cryo-ET densities in the Δ*flgV* flagellar basal bodies (Fig. 4F). To further demonstrate the association of this density with FlgV, we analyzed the basal bodies of *flgV-gfp* spirochetes. The FlgV C-terminal fusion to GFP increases the size of FlgV by 237 aa. We observed a significant increase in the FlgV-associated cryo-ET density for WT *flgV-gfp* samples (green arrows; Fig. 4G). These data support the model that FlgV localizes to the flagellar basal body between the C and MS rings of the flagellar rotor (Fig. 4C) as a ring of multiple FlgV molecules, and that the corresponding cryo-ET density is not part of the C-ring protein FliG2 as suggested previously[40]. We also determined the basal body structure when FlgV was overproduced (Fig. 4H). Surprisingly, we observed no significant difference in the WT/p$_{con}$ ++*flgV* basal body structures compared to those in WT/p, suggesting that FlgV overproduction did not alter the flagellar basal body structure.

## FlgV impacts the assembly of flagellar filaments

*B. burgdorferi* B31 standardly has 7–11 flagellar basal bodies at each cell pole, and each flagellar basal body is associated with one hook and a long filament that extends through the periplasmic space[42,43]. To understand the cell division and motility defects associated with altering FlgV levels, we used cryo-ET to visualize periplasmic flagella when *flgV* was deleted or the protein was overexpressed. As expected, in WT/p spirochetes we observed 7–11 flagellar filaments that formed a ribbon-like structure along the cell body (Fig. 5A; representative cell with 9 flagellar filaments). Δ*flgV*/p$_{ind}$ spirochetes had similar, but fewer, ribbon-like flagellar filaments, suggesting that the flagellar filaments are slightly shorter or reduced in number (Fig. 5B; representative cell with 7 flagellar filaments). *flgV* complementation restored the number of filaments to WT/p levels (Fig. 5C; representative cell with 9 flagellar filaments). Strikingly, when FlgV was overproduced fewer ribbon-like structures were observed, suggesting that the flagellar filaments are significantly shorter or reduced in number (Fig. 5D; representative cell with 4 flagellar filaments).

To further explore and quantify our observations, we used cryo-ET to examine the spirochete cell poles and enumerate both flagellar basal bodies and the associated filaments (Fig. 5E–H; representative tomograph projections for each strain; Supplementary Movies 1-4). In WT/p spirochetes, we observed equal numbers of basal bodies and filaments at the cell poles (Fig. 5E). In contrast, Δ*flgV*/p$_{ind}$ spirochetes had flagellar basal bodies with incomplete, short flagellar filaments (Fig. 5F) or basal bodies that lacked a filament entirely. Complementation restored the filaments to WT/p levels (Fig. 5G). In spirochetes overproducing FlgV, roughly half of flagellar basal bodies lacked filaments or had incomplete, short flagellar filaments (Fig. 5H). We observed no difference in the position of flagella across all *B. burgdorferi* strains examined. We counted the number of basal bodies

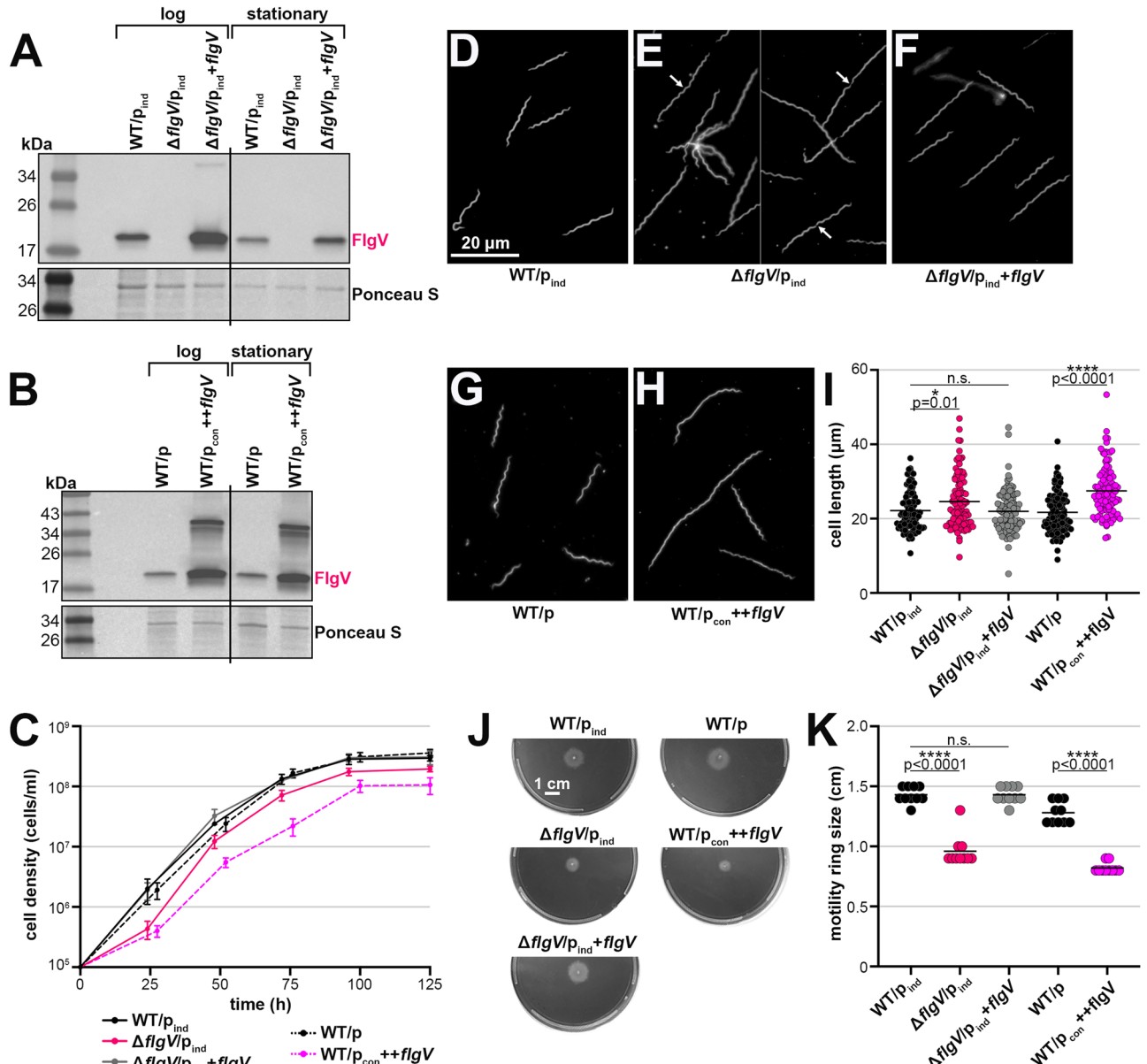

**Fig. 3 | FlgV levels impact *B. burgdorferi* cell division and motility.**
**A** Immunoblot analysis of FlgV levels in WT, *flgV* deletion and *flgV* complementa-tion strains. WT/$p_{ind}$ (PA273), Δ*flgV*/$p_{ind}$ (PA310), and Δ*flgV*/$p_{ind}$+*flgV* (PA312) were grown with 0.1 mM IPTG, after dilution of the starter culture, to an average density of ~1.1 × 10⁷ cells/ml (log) and ~2.3 × 10⁸ cells/ml (stationary) and total protein was isolated. Δ*flgV*/$p_{ind}$ (PA310) samples were collected 9 h after the WT/$p_{ind}$ (PA273) and Δ*flgV*/$p_{ind}$+*flgV* (PA312) samples, so that all cultures were collected at the same cell density. Protein extracts were separated on a Tris-Glycine gel, transferred to a membrane, stained with Ponceau S as a loading control, and probed with α-FlgV antibodies. Panel A was cropped to remove *B. burgdorferi* samples expressing *E. coli* Hfq. All samples are depicted in Fig. S4E. Size markers are indicated. **B** Immunoblot analysis of FlgV levels in WT and *flgV* overexpression strains. WT/p (PA023) and WT/$p_{con}$ ++*flgV* (PA267) were grown to an average density of ~2.6 × 10⁷ cells/ml (log) and ~1.8 × 10⁸ cells/ml (stationary) and total protein isolated. WT/$p_{con}$ ++*flgV* (PA267) samples were collected 6 h after the WT/p (PA023) sample, so that all cultures were collected at the same cell density. A dimer is detected with high levels of FlgV. Immunoblot was conducted as in panel (**A**). **C** Growth curve of *B. burgdorferi* expressing different levels of FlgV. Cell growth was monitored by dark field microscopy enumeration at the indicated time points, after dilution of the starter culture to 1 × 10⁵ cells/ml. Each data point (circles) represents the mean of three

biological replicates with the standard deviation. Dark field microscopy images of representative **D** WT/$p_{ind}$ (PA273), **E** Δ*flgV*/$p_{ind}$ (PA310), **F** Δ*flgV*/$p_{ind}$+*flgV* (PA312), **G** WT/p (PA023), and **H** WT/$p_{con}$ ++*flgV* (PA267). All cultures were grown to an average density of ~2.1 × 10⁷ cells/ml, washed with 1X PBS and imaged. For WT/$p_{ind}$, Δ*flgV*/$p_{ind}$, and Δ*flgV*/$p_{ind}$+*flgV*, 0.1 mM IPTG was added at the subculture. White arrow indicates septa. Scale bar on panel D applies to panels (**D**–**H**). **I** Quantification of spirochete length, for the strains in panels (**D**–**H**); *n* = 100. Cells were traced using the curve (spline) tool with ZEN 3.4 (blue edition) software. Each data point (circles) represents the length of one spirochete; the line corresponds to the mean length for each strain. Average length across WT/$p_{ind}$, Δ*flgV*/$p_{ind}$, and Δ*flgV*/$p_{ind}$+*flgV* samples or WT/p and WT/$p_{con}$ ++*flgV* were compared by one-way ANOVA with Tukey's multiple comparisons test or two-tailed t test, respectively, GraphPad Prism 9.5.1 (n.s., not significant). **J** Motility assay of *B. burgdorferi* expressing different levels of FlgV. Representative plates of each strain were photographed 9 d after inoculating into a BSKII 0.35% agarose plate. Scale bar on first plate applies to all plates. **K** Quantification of spirochete motility, for the strains in panel **J**. Each data point (circles; *n* = 10) represents the motility ring size (distance spirochetes spread from the inoculation site) for one sample; the line corresponds to the mean motility ring size for each strain. The statistical analysis was performed as reported for panel (**I**).

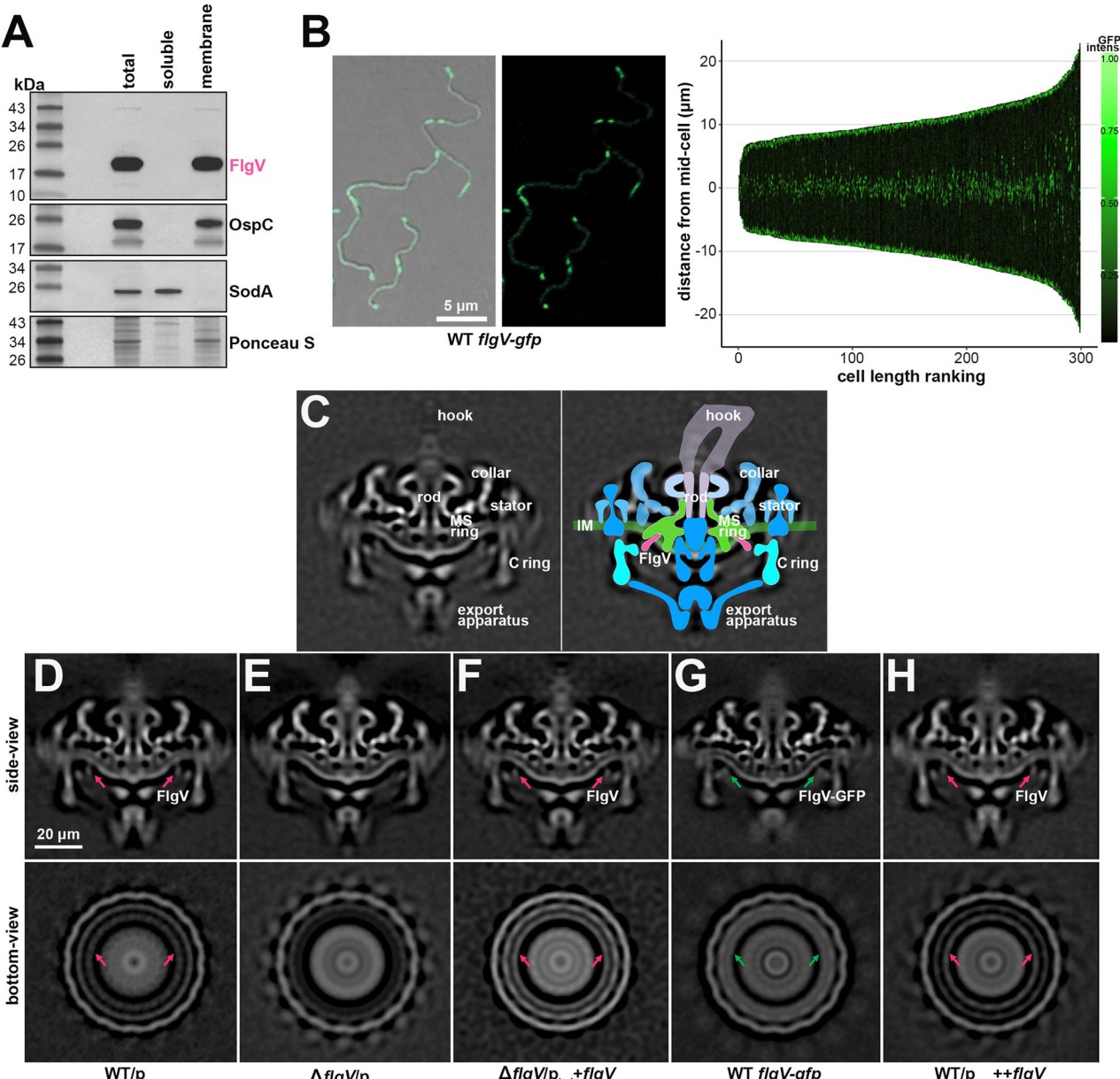

**Fig. 4 | FlgV is localized to the flagellar basal body. A** Immunoblot analysis of *B. burgdorferi* total protein lysate fractionated into soluble and membrane samples. WT *B. burgdorferi* (PA001) were grown to a density of ~2.2x10⁷ cells/ml and then lysed by sonication, total protein was isolated, and the samples were fractioned by ultracentrifugation. Protein extracts were separated on a Tris-Glycine gel, transferred to a membrane, stained with Ponceau S as a loading control, and probed with α-FlgV, α-OspC, and α-SodA antibodies. Proteins were probed sequentially on the same membrane; size markers are indicated. **B** Localization of FlgV-GFP by confocal microscopy. *B. burgdorferi* (PA402) were grown to a density of ~2.0x10⁷ cells/ml, prior to imaging. Left panels: Bright field and fluorescence composite (left micrograph) and fluorescence-only (right micrograph) are shown. Scale bar on left micrograph also applies to right micrograph. Right panel: Demograph of the GFP profile for 300 spirochetes, cells were positioned in order of increasing length.

**C** Reconstructed cryo-ET central (side-view) cross section of WT/p (PA023) *B. burgdorferi* flagellar basal body (left) with overlaid cartoon model of the flagellar basal body (right). The inner membrane (IM, green), export apparatus (blue), C ring (light blue), MS ring (green), rod (lavender), hook (light purple) and FlgV (dark pink) are labeled. Reconstructed central (side-view) and bottom-view cryo-ET cross sections of **D** WT/p (PA023; repeated from panel C), **E** Δ*flgV*/p$_{ind}$ (PA310), **F** Δ*flgV*/p$_{ind}$+*flgV* (PA312), **G** WT *flgV-gfp* (PA402), and **H** WT/p$_{con}$ ++*flgV* (PA267) *B. burgdorferi*. All cultures were grown to an average density of ~3.6 × 10⁷ cells/ml, washed with 1X PBS, mixed with 10 nm gold particles, deposited on glow-discharged grids and imaged. Tomographs were generated, aligned and reconstructed for each strain. Densities predicted to correlate with FlgV (pink arrow) and FlgV-GFP (green arrow) are indicated. Scale bar on panel D applies to panels D–H.

and filaments at one cell pole for ~20 cells per strain. WT/p, Δ*flgV*/p$_{ind}$, and Δ*flgV*/p$_{ind}$+*flgV* cells all had similar numbers of flagellar basal bodies (Fig. 5I; mean values of 8.2 ($n$ = 18), 7.7 ($n$ = 21) and 8.4 ($n$ = 18), respectively). However, there was a significant reduction in the average number of flagellar filaments for Δ*flgV*/p$_{ind}$, mean value of 6.7 ($n$ = 21), compared to WT/p, 8.2 ($n$ = 18) and the *flgV* complemented spirochetes, 8.5 ($n$ = 18), in which the average number of basal bodies and

filaments were the same. In a separate experiment, we tallied the flagellar basal bodies and filaments in spirochetes when *flgV* was overexpressed (Fig. 5J). There were some variations in the numbers for WT/p$_{ind}$ and WT/p *B. burgdorferi* between experiments (Fig. 5I, J), possibly due to differences in the precise cell density of the sampled culture. Nevertheless, there consistently was a reduction in the average number of flagellar basal bodies in *B. burgdorferi* overexpressing *flgV*, with a

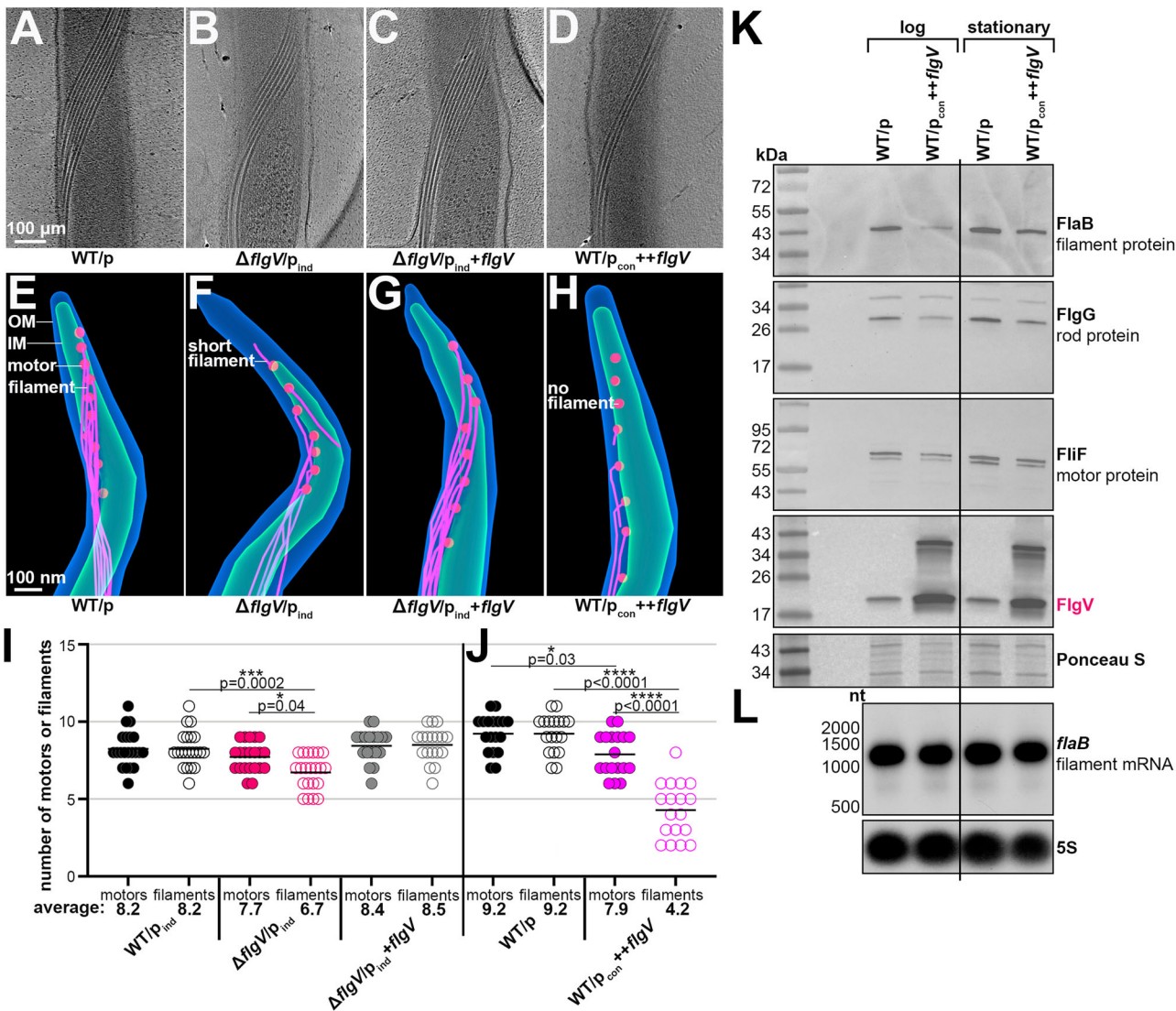

**Fig. 5 | FlgV levels impact the assembly of flagellar filaments.** Surface-view cryo-ET micrograph of representative central periplasmic cell sections of **A** WT/p (PA023), **B** Δ*flgV*/p_ind (PA310), **C** Δ*flgV*/p_ind+*flgV* (PA312), and **D** WT/p_con + +*flgV* (PA267) *B. burgdorferi*. All cultures were grown to an average density of ~3.6 × 10^7 cells/ml, washed with 1X PBS, mixed with 10 nm gold particles, deposited on glow-discharged grids, and imaged. Scale bar on panel A applies to panels (**A**–**D**). **E**–**H** Projection images of a representative cell pole for each strain, described in panels A–D. The outer membrane (blue), inner membrane (green), flagellar basal bodies (red) and flagellar hooks and filaments (pink), were manually segmented in IMOD and are indicated. Scale bar on panel E applies to panels E–H. See Supplementary Movies 1-4. **I**–**J** Quantification of flagellar basal bodies and filaments. Individual spirochetes were tallied for number of basal bodies and filaments at one cell pole. Each data point (circles) represents one spirochete (*n* = 18-21); the line corresponds to the mean number of flagellar basal bodies or filaments for each strain. Panels I and J were independent experiments; panel I: 0.1 mM IPTG was added at the subculture and WT/p_ind (PA273), Δ*flgV*/p_ind (PA310), and Δ*flgV*/p_ind+*flgV* (PA312) cultures were grown to an average density of 3.8 × 10^6 cells/ml; panel J: WT/p (PA023) and WT/p_con + +*flgV* (PA267) were grown to an average

density of 2.3 × 10^7 cells/ml. Average flagellar basal bodies and filament numbers per sample were compared by one-way ANOVA with Tukey's multiple comparisons test, GraphPad Prism 9.5.1. **K** Immunoblot analysis of flagellar structural proteins when FlgV is overexpressed. The membrane from Fig. 3B was stripped and reprobed with α-FlaB[74], α-FlgG and α-FliF antibodies. The FlgV panel was repeated from Fig. 3B; the Ponceau S panel is the same as Fig. 3B, but shows a different region; size markers are indicated. The fold change (FlgV overexpression compared to WT) of specific bands was calculated by densitometry analysis: FlaB, log: 1.2 fold decrease; FlgG, log: 1.2 fold decrease; FliF, log: 1.0 fold decrease; FlgV, log: 2.7 fold increase; and FlaB, stationary: 1.3 fold decrease; FlgG, stationary: 1.3 fold decrease; FliF, stationary 1.1 fold decrease; FlgV, stationary: 1.9 fold increase (Image J 1.53t; normalized to the nonspecific band detected with the α-FlgG antibody). A second independent immunoblot was performed (Fig. S8C). **L** Northern analysis of *flaB* levels. Total RNA was isolated from a portion of the same cultures used in panel K. RNA was separated on an agarose gel, transferred to a membrane and probed for *flaB*. The membrane was stripped and probed for 5S as a loading control; size markers are indicated.

mean value of 7.9 (*n* = 18), compared to WT/p spirochete basal bodies, mean value of 9.2 (*n* = 21). More strikingly, there was a significant reduction in the average number of flagellar filaments in spirochetes overproducing FlgV protein, mean value of 4.2 (*n* = 18), lower than the number of flagellar basal bodies in the overexpression strain and lower than the number of flagellar filaments in WT/p spirochetes, mean value of 9.2 (*n* = 21). Collectively, our data provide direct evidence that

changes in FlgV levels have profound impacts on the assembly of the flagellar filament, while they have limited impacts on the structure, number, and position of flagellar basal bodies.

To further investigate the impact of *flgV* overexpression, we examined the total protein levels for specific components of the flagellar basal body and filament. In *B. burgdorferi*, the flagellar filament is comprised of the major flagellin protein, FlaB, and a minor sheath

protein, FlaA[44,45]. Compared to WT/p samples, FlaB levels decreased in lysates isolated from WT/p_con ++*flgV B. burgdorferi* at both logarithmic and stationary phase (Fig. 5K, see figure legend for quantification by densitometry). We also observed a ++*flgV*-dependent decrease in FlgG, a flagellar rod protein which connects the flagellar filament to the basal body. The levels of the motor protein FliF also decreased with *flgV* overexpression. These FlgV-effects were more prominent in the logarithmic phase samples, compared to the stationary phase samples. We also isolated and analyzed RNA from WT/p and WT/p_con ++*flgV B. burgdorferi* from the same experiment. Northern analysis indicated no change in the levels of *flaB* mRNA when *flgV* is overexpressed (Fig. 5L). These data support a model that FlgV impacts flagellar assembly post-transcriptionally. We also examined FlaB levels when *flgV* was deleted; no significant decrease in FlaB protein (Fig. S8A) or mRNA levels (Fig. S8B) were observed. The same conclusions were drawn from a second, independent experiment (Fig. S8C).

Collectively, our findings implicate FlgV as an important component of the flagellar basal body that impacts the late-stage assembly of flagellar filaments. Given this effect on the flagella, we hypothesized that FlgV could play a role in controlling spirochete motility during the tick–mammal enzootic cycle.

### *flgV* is dispensable for *B. burgdorferi* survival and replication in ticks

*B. burgdorferi* have been observed to be non-motile in the midguts of infected unfed ticks but become motile during the bloodmeal[11]. To determine the importance of *flgV* throughout the enzootic cycle of *B. burgdorferi*, we artificially infected larval ticks by submerging pools of naïve *Ixodes scapularis* ticks in BSKII medium[46] containing WT/p, Δ*flgV*/p or WT/p_con ++*flgV* spirochetes and monitored the infection at each stage of tick-feeding on mice (Fig. 6A). A subset of artificially-infected, unfed larval ticks were assayed for *B. burgdorferi* burden by crushing and plating for colony-forming units (CFUs) in solid BSKII medium[47]. All strains colonized the unfed larvae at the same level, following the immersion procedure, confirming successful artificial infection (Fig. 6B).

To test the ability of Δ*flgV* or ++*flgV* spirochetes to replicate in feeding ticks, infected larvae were fed to repletion (allowed to naturally fall off) on naïve mice and collected for analysis. The spirochete burden for individual fed larval ticks was determined by enumeration of CFUs. Not all ticks acquire *B. burgdorferi* by immersion infection[46], and therefore, it is important to quantify the initial infection efficiency for each bacterial strain. The larval tick-infection efficiency of WT *B. burgdorferi* was calculated at 35% (6 infection positive ticks/17 total assayed ticks), which was similar to larvae infected with Δ*flgV*/p (25%; 4/16 ticks) or WT/p_con ++*flgV* (25%; 5/20 ticks) (Fig. 6C; reported below *x*-axis). We then quantified and compared the spirochete burdens for only the infected ticks in each group. There was no significant difference in the average number of *B. burgdorferi* per tick when *flgV* was deleted or FlgV protein was overproduced, compared to ticks infected with WT bacteria (Fig. 6C).

Fed larvae were allowed to molt into nymphs and assessed for *B. burgdorferi* infection post-molt. There was no difference in spirochete survival through the molt for the *flgV* strains, compared to WT spirochetes (Fig. 6D). Nymphs were fed to repletion on naïve mice and again, there was no significant difference in the average number of spirochetes per tick for any *B. burgdorferi* strain (Fig. 6E). These data document that *flgV* is dispensable for *B. burgdorferi* survival and replication in larval and nymphal ticks.

We also determined the ability of the *B. burgdorferi* strains to infect mice by reisolating spirochetes from tissues approximately three weeks after the larval tick bite (Fig. 6F). WT spirochetes were reisolated from the ear, heart, bladder and joint tissues of all three mice. In contrast, Δ*flgV*/p spirochetes were detected in all tissues of only two out of three mice and there was no reisolation of *flgV*

overexpression spirochetes from any mouse tissue. In the nymphal tick feeding, we did not collect enough nymphs per mouse to confidently assess defects in mouse infectivity. Our data show that *flgV* is important for efficient *B. burgdorferi*-mammalian infection by tick bite and led us to further examine the consequences of altering FlgV protein levels on the ability of *B. burgdorferi* to infect mice.

### *flgV* is critical for *B. burgdorferi* dissemination and infectivity in mice

Aspects of tick bite transmission of *B. burgdorferi* can be modeled by intradermal needle inoculation of mice (Fig. 7A). Successful mammalian infection by *B. burgdorferi* begins with intradermal inoculation of spirochetes, which is followed by localized replication of bacteria at the inoculation site[48,49]. At 5 to 7 days after the intradermal needle injection, spirochetes disseminate and are detectable in the bloodstream, with a peak in bacteremia at day 6[50]. At 7 to 10 days after inoculation, spirochetes exit the blood and start to be detectable in distal tissues, such as secondary skin sites and ear, heart, and joint tissues, where long-term colonization is established[48,49].

Using this model, we evaluated the contribution of *flgV* at each stage of mouse infection by quantifying spirochete loads and reisolating spirochetes from tissues. Groups of mice were intradermally inoculated with WT and Δ*flgV* spirochetes and infection parameters were assessed over a three-week infection time course. We first confirmed that equal number of WT and Δ*flgV* spirochetes were inoculated into the skin 1 hour after intradermal delivery by extracting DNA and quantifying the number of *B. burgdorferi* by qPCR (Fig. 7B). There was no significant difference between WT or Δ*flgV* spirochete burden in the skin inoculation sites at the 1-hour timepoint. In parallel, using a section of the same skin sample examined by qPCR, we assayed the ability to reisolate WT or Δ*flgV* spirochetes from the inoculation site in liquid medium (Fig. 7B; reported below the *x*-axis). Again, we observed no significant difference between the two strains. As expected, the numbers of WT spirochetes expanded in the skin inoculation site from 1 hour to 7 days post-infection. We were able to reisolate both WT and Δ*flgV* spirochetes from the inoculation site of all mice at the 3- and 7-day timepoints. However, compared to WT spirochetes, there were significantly fewer Δ*flgV* spirochetes in the skin inoculation sites at 3 and 7 days. Our data indicate that *flgV* is important for maximum *B. burgdorferi* expansion in the skin, during the first 7 days of infection.

Moving away from the site of injection, we examined the spirochete load in blood during hematogenous dissemination in mice by measuring spirochete bacteremia. Following an intradermal needle injection, WT *B. burgdorferi* are only detectable in blood approximately 5–7 days post-inoculation[50]. Blood was collected from mice at days 6, 7, and 8 post-injection and plated for *B. burgdorferi* CFUs (Fig. 7C). As expected, WT *B. burgdorferi* were detected in blood at day 6, with a decrease in numbers at day 7 and below the limit of detection by day 8. Strikingly, no Δ*flgV* spirochetes were detected at day 6 and only two mice had detectable Δ*flgV* spirochetes in blood at day 7. Therefore, Δ*flgV* spirochetes may be delayed in entering the bloodstream, be at low numbers during hematogenous spread and/or be rapidly eliminated from the blood, ultimately attenuating spirochete dissemination to distal tissues.

We finally assessed the presence and amount of disseminated spirochetes in distal tissues. The initial skin inoculation site and distal tissues: ear, heart, and joint, were harvested and subjected to reisolation and qPCR analysis at 14 and 21 days post-inoculation (Fig. 7D). At the 14-day timepoint, all tissues (24/24 of total skin, ear, heart and joint tissues) from the mice infected with WT *B. burgdorferi* were reisolation positive with quantifiable spirochetes. Only 37.5% (9/24) of mouse tissues were reisolation positive for Δ*flgV B. burgdorferi* at day 14 (Fig. 7D; reported for each tissue below the *x*-axis). The numbers of Δ*flgV* spirochetes in tissues were generally below the limit of detection by qPCR at the 14-day timepoint. No Δ*flgV* spirochetes were detected in

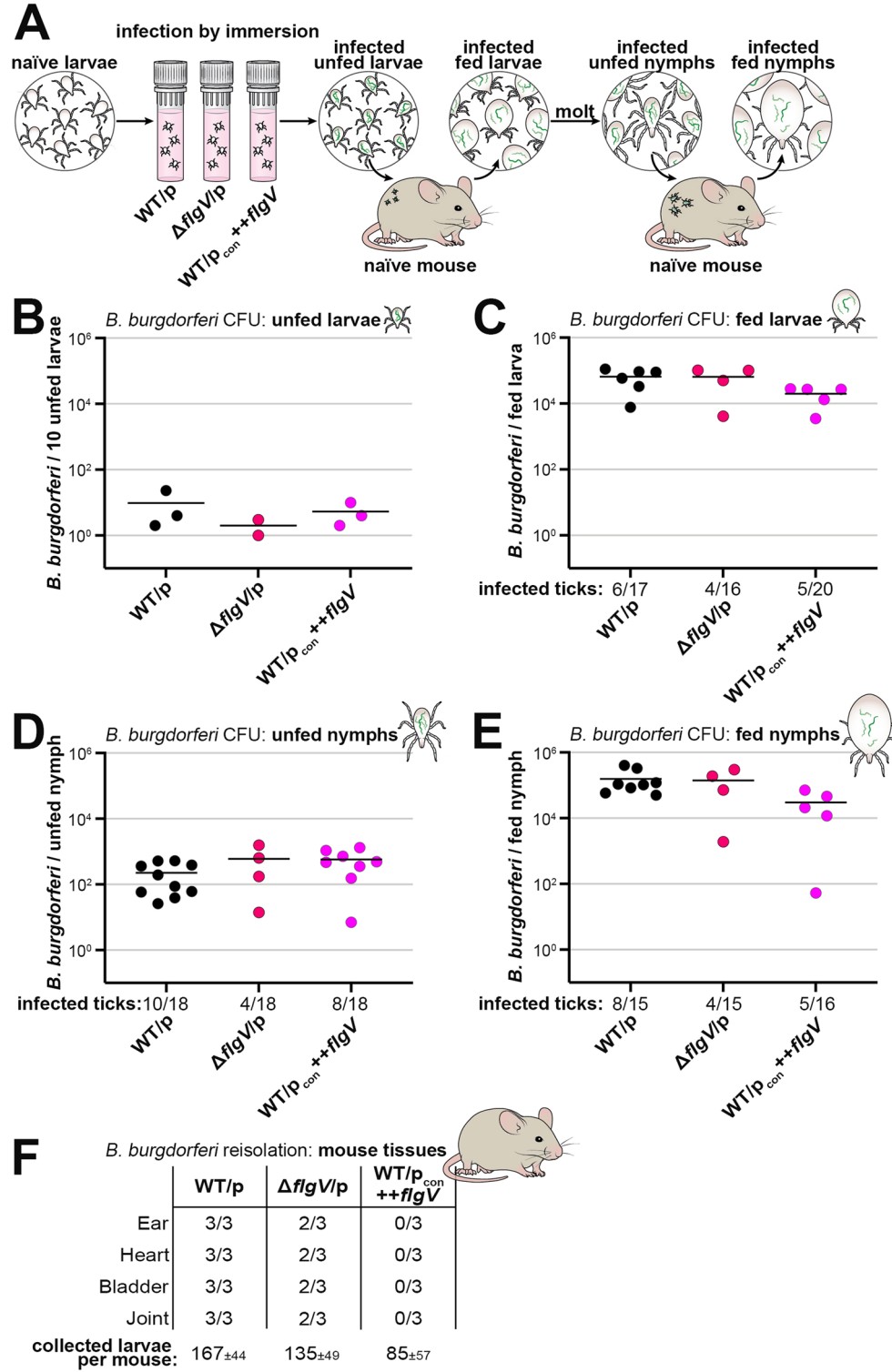

**Fig. 6 | *flgV* is not important for *B. burgdorferi* replication or survival in *Ixodes* ticks. A** Schematic of artificial tick infection by immersion used for tick-mouse infection study. Naïve, unfed larvae were desiccated and submerged in BSKII containing WT/p (PA023), Δ*flgV*/p (PA257), or WT/p_con ++*flgV* (PA267) *B. burgdorferi*. Ticks were assayed for infectivity at each life-stage: **B** unfed larvae, **C** fed larvae, **D** unfed nymphs, and **E** fed nymphs. Ticks were surface sterilized, crushed and plated in solid BSKII containing RPA cocktail and gentamycin. Data points (circles) represent the number of colony-forming units (CFUs) for infected ticks. For unfed larvae (panel **B**), 3 groups of 10 unfed larvae from each strain were assayed for infectivity; one plate of the Δ*flgV*/p sample was contaminated and not

able to be counted. Ticks from all other life-stages (panels **C**–**E**), were assayed individually for infectivity; the number of individual infected ticks over the number of individual crushed ticks is listed below the *x*-axis. The average numbers of *B. burgdorferi* per tick, for each tick life-stage, were compared by one-way ANOVA with Tukey's multiple comparisons test, GraphPad Prism 9.5.1; no significant difference was observed for any tick life-stage. **F** Reisolation of *B. burgdorferi* from mouse tissues after larval tick feeding. 3 mice per group were assessed for infection 3 weeks post larval tick-feeding. Ear, heart, bladder and joint tissues were collected and analyzed for spirochete reisolation in BSKII by dark-field microscopy enumeration. Average numbers of larval ticks collected per mouse are indicated.

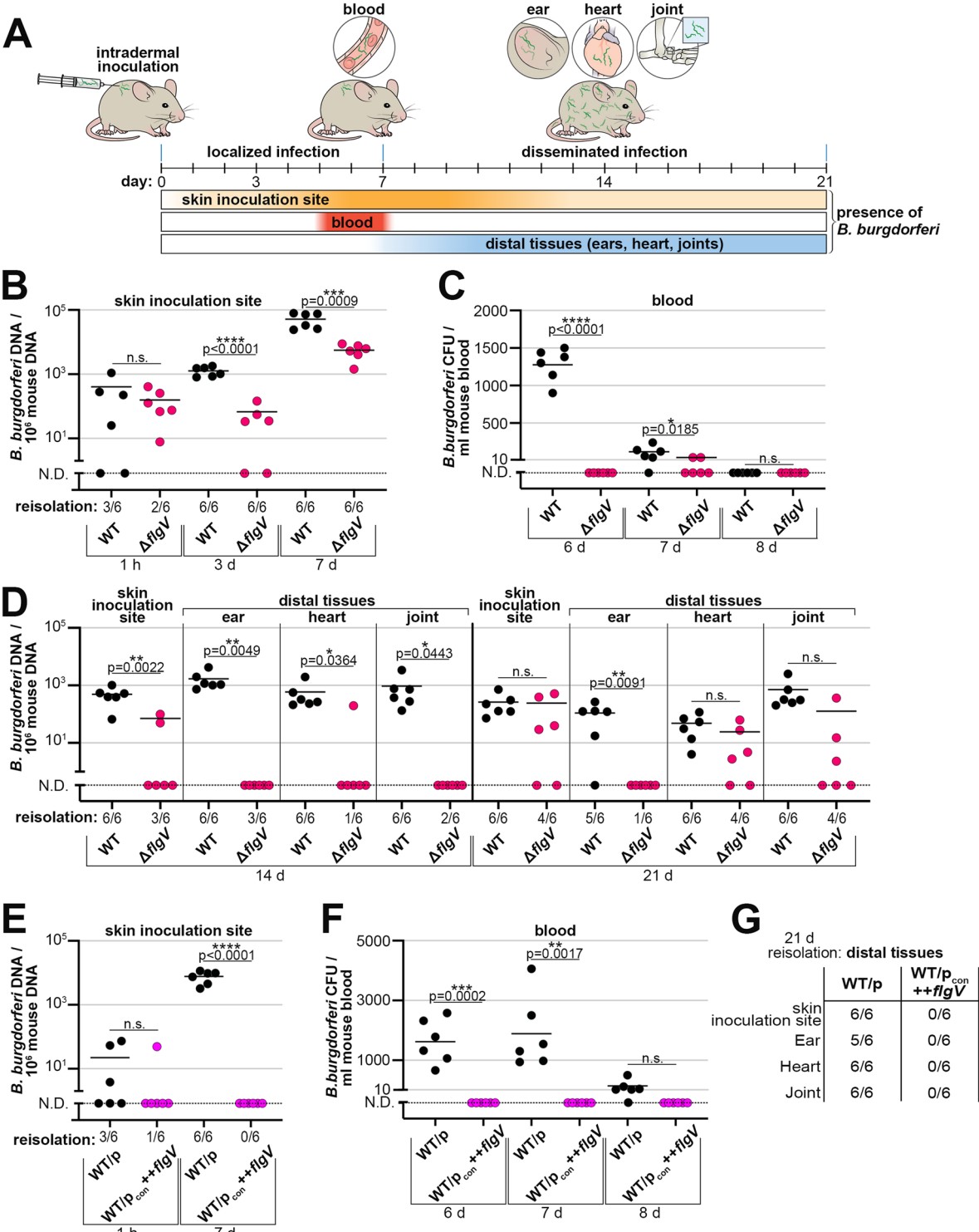

**Fig. 7 | *flgV* is important for timely and productive dissemination in mice.**
**A** Schematic of mouse-*B. burgdorferi* infection kinetics by intradermal needle inoculation. Mice (*n* = 6) were infected intradermally with 1 × 10⁴ WT (PA001) or Δ*flgV* (PA251) *B. burgdorferi*. After the indicated times, the **B** skin inoculation site, **C** blood, and **D** distal tissues were assessed for spirochete infectivity and burden. In a separate experiment, mice were infected intradermally with 1 × 10⁴ WT/p (PA023) or WT/p$_{con}$ + +*flgV* (PA267) *B. burgdorferi*. After the indicated times, the **E** skin inoculation site, **F** blood, and **G** distal tissues assessed for the presence of spirochetes. For panels B–G: Each data point (circles) represents one mouse; samples with no detectable spirochetes are indicated by a data point on the dotted line (N.D. = not detected). The

mean for each strain is indicated by a line, only detected data were used to calculate the mean. For quantitative-PCR analysis, DNA was extracted from tissues and *B. burgdorferi* load was measured by quantifying *B. burgdorferi flaB* copies normalized to 10⁶ mouse *nid* copies, in technical triplicate for each sample. For bacteremia analysis, blood was collected, diluted in BSKII and plated. For tissue reisolation, skin inoculation site, ear, heart, and joint tissues were collected, submerged in BSKII and enumerated by dark-field microscopy. The calculated means between WT versus Δ*flgV* or WT/p versus WT/p$_{con}$ ++*flgV* for each tissue/blood at a given timepoint were compared by a one-tailed, unpaired t test; for all statistical analyses N.D. data points were included as zero; GraphPad Prism 9.5.1; (n.s., not significant).

ear and joint tissues, and only one heart and two skin samples had detectable Δ*flgV* spirochetes. At 21 days-post inoculation, again WT spirochetes were reisolated and quantified from all tissues, excluding one ear sample (23/24). For the Δ*flgV* *B. burgdorferi* infection, 54.2% (13/24) of mouse skin, heart and joint samples were reisolation positive at day 21 (Fig. 7D; reported below the *x*-axis). Despite low dissemination and infection rates, quantification of these tissues revealed no significant difference between the numbers of detected WT and Δ*flgV* spirochetes. These data suggest that if a tissue becomes infected with Δ*flgV* *B. burgdorferi*, the spirochetes ultimately can reach a similar level of tissue colonization, compared to WT *B. burgdorferi*. In contrast, and strikingly, only one mouse ear was reisolation positive for Δ*flgV* spirochetes and no Δ*flgV* spirochetes were detected, above the limit of detection, in any ear sample at day 21. Overall, we detected few Δ*flgV* spirochetes at 14 days post inoculation, but at 21 days post inoculation Δ*flgV* spirochetes were detected in a majority of mice. Our data suggests a *flgV*-dependent delay in the kinetics of *B. burgdorferi* dissemination and infection.

We also assayed mice that were intradermally inoculated with *B. burgdorferi* overproducing FlgV protein. Remarkably, no WT/p$_{con}$ + +*flgV* spirochetes were detected in the skin inoculation sites by qPCR or tissue reisolation at 7 days post inoculation (Fig. 7E). Spirochetes overproducing FlgV protein were also not detected in the blood at 6-, 7- or 8-days post needle injection (Fig. 7F). Finally, no WT/p$_{con}$ ++*flgV* spirochetes were able to be reisolated from any mouse tissue 21 days after the intradermal inoculation (Fig. 7G). In contrast, WT/p spirochetes were robustly detected in the inoculation site (Fig. 7E), in the blood (Fig. 7F) and in distal tissues (Fig. 7G). These data, consistent with our previous observations, indicate that increased levels of FlgV have greater consequences than *flgV* deletion on flagellar filaments and *B. burgdorferi* infection of the mammalian host.

Collectively, our analysis of key timepoints of *B. burgdorferi* infection of mice documented that FlgV, and thus proper flagellar filament synthesis, is critical for timely and productive dissemination during an infection model of Lyme disease.

## Discussion

Here we report a novel component of *B. burgdorferi* flagellar basal bodies that affects flagellar assembly, and thus significantly impacts motility and dissemination during mammalian infection. A previous study reported that *bb0268* (*flgV*) encodes an atypical Hfq protein[19]. Thus, we set out to characterize the RNAs associated with BB0268, but in the absence of identifying any BB0268-bound RNAs, discovered *bb0268* was misannotated.

### BB0268 (FlgV) is not an Hfq homolog or an RNA-binding protein

RNA-binding proteins have been shown to play critical roles in modulating gene expression and are required for infection across a broad range of bacteria (reviewed in ref. 51). These RNA-binding proteins can facilitate sRNA–RNA interactions, like the Hfq and ProQ proteins (reviewed in refs. 20,21,27) or primarily bind mRNAs, such as CsrA (reviewed in ref. 52), with new classes in bacteria emerging, such as KH domain proteins (reviewed in ref. 24).

A co-IP experiment is typically used to characterize RNA-binding proteins and identify bound RNAs. None of our co-IP experiments showed that BB0268 (FlgV) binds RNA, despite strong enrichment of both 3XFLAG-tagged and native FlgV protein in independent experiments. We analyzed the conservation of Hfq in bacteria and found no Hfq homologs in *B. burgdorferi*. Additionally, we were unable to reproduce previous results[19] that claimed *flgV* complements *E. coli* Δ*hfq* phenotypes. We also assayed the effect of expressing *E. coli hfq* in Δ*flgV* *B. burgdorferi* and found no complementation of the *flgV*-dependent defects in growth (Fig. S4D and S4E). Caution generally should be taken when interpreting results from ectopic gene expression in disparate bacteria. Another study misannotated an RNA binding

protein in *Pseudomonas aeruginosa* because the purified protein samples from *E. coli* extracts were contaminated with *E. coli* Hfq[53]. Stress responses or other indirect effects also may be invoked from the expression of a foreign protein in a bacterium.

The effect of *flgV* on the RpoS regulator was examined both in *E. coli* and *B. burgdorferi*. Across bacteria, the activation of RpoS is complex, as it can involve multiple stress signals and is controlled at the level of transcription, translation and RNA/protein stability (reviewed in ref. 30). In *E. coli*, we observed some minor effects on RpoS levels when *flgV* was ectopically expressed; however, these outcomes could be indirect. In *B. burgdorferi*, RpoS is a cornerstone for responding to environmental changes during the enzootic cycle, and therefore, its regulation must be tightly controlled[54]. It was previously proposed that FlgV post-transcriptionally regulates *rpoS* because it was reported that deletion of *flgV* led to higher levels of *rpoS* mRNA, but lower levels of RpoS protein and lower levels the RpoS-regulated *ospC* mRNA[19]. However, we found that when *rpoS* levels increased in Δ*flgV* samples there was a concomitant increase in RpoS protein, *ospC* mRNA and OspC protein (Fig. S8A and S8B), opposite to what was reported previously[19]. There were some variations in the *flgV*-*rpoS* effect between experiments (Fig. S6B), which could be due to the precise state of the collected cells. Even across four WT *B. burgdorferi* samples, which all had similar cell densities by darkfield microscopy enumeration, there were some variations in *rpoS* levels (Fig. S8B). None of our observations support a hypothesis[19] that FlgV post-transcriptionally regulates RpoS.

Given that *B. burgdorferi* has a limited number of characterized transcription factors[55], RNA-based regulation is an attractive mechanism for fine-tuning gene expression during the transmission and survival between tick and mammalian environments. Thus, future work is warranted to identify and properly characterize bona fide RNA chaperones in *B. burgdorferi*.

### FlgV is a broadly conserved flagellar protein

In Spirochaetota, endoflagella are critical for cell shape, motility and survival. For *B. burgdorferi*, spirochetes in the *Ixodes* tick midgut must sense a blood meal and migrate into a mammalian host and, likewise, *B. burgdorferi* inhabiting a mammal sense a feeding tick and migrate into that tick. Thus, to effectively maintain the enzootic cycle, the *B. burgdorferi* gene products which assemble, propel, and break flagellar motors must be tightly controlled. The regulatory mechanisms that control *B. burgdorferi* flagellar assembly and function are largely unknown and, intriguingly, the transcription of *B. burgdorferi* flagellar biosynthesis genes has been reported to be constitutive. We showed that *B. burgdorferi flgV* is transcribed as part of complex transcriptional units with surrounding genes in the "*flgB* superoperon". The numerous and overlapping RNA transcripts in this region may have confounded the interpretations of experiments using reverse transcriptase-qPCR across ORF junctions[17]. Complex transcription, mRNA processing, and the presence of sRNAs across this operon may serve regulatory functions for the expression of specific genes, and thus control of motility, during the enzootic cycle.

There is an inherent link between the cell-cycle and flagella integrity for spirochetes[6,38,56], and we observed that when *flgV* is missing, spirochetes have defects in cell division and motility. Flagella-mediated motility is likely required to efficiently separate dividing spirochetes[6,38,56]. We detected an abundance of conjoined spirochetes, and the presence of septa, for logarithmically growing Δ*flgV* *B. burgdorferi* in culture. In stationary phase, spirochetes lacking *flgV* were the same length as WT spirochetes. This suggests that in the absence of *flgV*, cells have prolonged cell division when rapidly growing, although ultimately division can take place. Cell elongation was also observed with FlgV overproduction in logarithmic-phase spirochetes. Cell division genes (*ftsZ* (*bb0299*), *ftsA* (*bb0300*), *ftsW* (*bb0302*)) are encoded upstream of the "*flgB* superoperon". Cross-talk between flagellar

assembly and cell division has also been observed in other bacteria. Cell division-site placement in *Campylobacter jejuni* is influenced by *flhG*, the gene encoding a regulator of flagella number[57]. As in *B. burgdorferi*, *C. jejuni flhG* is encoded directly upstream of the *C. jejuni flgV* homolog. In *Caulobacter crescentus*, cell division is regulated in tandem with flagellar assembly[58]. Further work is warranted to explore the connection between cell division proteins and motility in *B. burgdorferi*.

We found *flgV* contributes to *B. burgdorferi* flagellar assembly. Δ*flgV* spirochetes have shorter and fewer flagellar filaments, compared to WT spirochetes. Yet strikingly, the number of flagellar basal bodies is not impacted in Δ*flgV* spirochetes, nor is the overall structure of the flagellar basal body. Five other *B. burgdorferi* mutants have demonstrated a similar flagellar filament defect. Deletions of *fliH* or *fliI*, encoding components of the flagellin export apparatus, result in short flagellar filaments but do not prevent the formation of flagellar basal bodies and filaments[38]. Deletion of *fliD*, encoding the flagellar cap protein, deletion of *flaB*, encoding the major flagellin protein, or deletion of *flgJ*, encoding a protein that impacts flagellar hook assembly, decreases/abolishes flagellar filaments and results in short, disoriented flagellar hooks[59,60].

Our study used cryo-ET to document that FlgV is an inner membrane protein and that multiple FlgV molecules form a ring at the junction between the MS- and C-rings of the flagellar basal body. Our data set the stage for further exploration of FlgV function and the control of motility. In general, once the flagellar basal body is assembled, flagellin is secreted to build the filament[61,62]. Therefore, flagellin homeostasis and export of cytosolic flagellin into the basal body generally is co-regulated[63,64]. Typically, in model Enterobacteriaceae and Bacillaceae, $\sigma^{28}$ controls the transcription of genes encoding the flagellar filament. *Borreliae* lack $\sigma^{28}$. In this study, we showed that FlgV, which is strongly co-conserved with $\sigma^{28}$ across other Spirochaeta and Firmicutes and other phyla, is another protein that modulates flagellar filament synthesis.

The stoichiometry of flagellar proteins is likely crucial for proper flagellar assembly. Yet, surprisingly, we observed no noticeable change to the flagellar basal body structure when FlgV was overproduced, despite severe impacts to flagellar filamentation. FlgV harbors a conspicuous, conserved intramembrane basic residue (in an "RS" amino acid signature in *Borrelia*) in the second transmembrane helix, hinting that the protein could act as an intramembrane flagellar sensor (sensing a membrane signal through these polar residues). We hypothesize that environmental signals may promote/inhibit FlgV activity, which might be to tether/bring factors to the flagellar apparatus, perhaps through protein–protein associations at the FlgV C-terminal peptide tail.

We observed three distinct conservation patterns of FlgV in bacteria. We defined these as Type-1 in the Spirochaetota, PVC clade, Firmicutes, Nitrospinae, and several other less-studied bacterial phyla, Type-2 in the *Helicobacter*- and *Campylobacter*-like Epsilonproteobacteria, and Type-3 in the *Arcobacter*-like Epsilonproteobacteria. Notably, the Epsilonproteobacteria (Type-2 and Type-3) FlgV differ from the Type-1 FlgV in that they lack: (i) the characteristic arginine in the second TM helix; (ii) an acidic residue at the start of TM helix-2; and (iii) a conserved polar residue towards the C-terminus of TM helix-2. Keeping with these differences, the Type-2 and Type-3 FlgV did not retrieve Type-1 FlgV with strong statistical significance. Nevertheless, structural comparisons revealed that they adopted a similar structure and they showed comparable gene-neighborhood associations within the flagellar operons. Simultaneous to our work, another group showed FlgV in *Helicobacter pylori* also associates with flagellar basal bodies and impacts motility[65]. However, there were clear differences between the FlgV in *B. burgdorferi* and FlgV in *H. pylori*. The FlgV ring in *H. pylori*, observed by cryo-ET has large and wide densities[65]. A previous study in *Campylobacter jejuni*[35] reported FlgV interacts with additional flagellar proteins. We did not identify a FlgV homolog in flagellated-Enterobacteriaceae, all of which harbor $\sigma^{28}$. Yet, there might be other functional analogs of FlgV that were not found by our analysis. The mechanistic details of how FlgV impacts flagellar filamentation, possible FlgV interacting partners, and the differences across bacterial species are exciting directions for future work.

While endoflagella define spirochetes, there are notable differences (reviewed in ref. [5]). *Borrelia*, which rely on a tick vector for survival, uniquely lack $\sigma^{28}$. Constitutive expression of flagellar genes may be required for a spirochete to quickly respond to a sporadic bloodmeal and migrate into a mammalian host. Other forms of regulation including the modulation of flagellar filamentation, perhaps through *flgV*, is likely critical for energy conservation and increased motility during a bloodmeal. A limited number of flagella are required for maintaining cell shape in *Borrelia*, but there may be times during the enzootic cycle such as in an unfed tick, where some flagellar basal bodies lack a filament. Future experiments to examine FlgV and other proteins encoded by the heterogeneous flagellar superoperons likely will reveal new insights into the regulation of these important nanomachines, which impact motility, morphology, cell division, and infectivity across the bacterial superkingdom.

## Importance of *B. burgdorferi* motility during infection
Pathogens must overcome numerous physical conditions, nutrient limitations and immune defenses to survive inside a host. *B. burgdorferi*, with a small, segmented genome and no known virulence factors, heavily relies on motility to move from host-to-host, scavenge for nutrients and avoid immune factors. We undertook a unique opportunity to investigate the importance of motility during infection by testing Δ*flgV* and ++*flgV* spirochete survival during tick and mammalian infection. On average, we found WT spirochetes to harbor 9 filaments, Δ*flgV* spirochetes to harbor 7 filaments, and ++*flgV* spirochete to harbor 3 filaments. Both Δ*flgV* and ++*flgV B. burgdorferi* survived and replicated in ticks at the same levels as WT *B. burgdorferi*. This suggests that full motility is not critical for *B. burgdorferi* inside ticks. A previous study observed that *B. burgdorferi* lacking flagellar filaments entirely (Δ*flaB*) can survive, but at a reduced burden, in ticks[9]. On the other hand, Δ*flgV* and ++*flgV B. burgdorferi* were attenuated in their ability to infect mice.

Successful mammalian infection of *B. burgdorferi* begins with a localized skin infection of spirochetes at the tick bite site, followed by dissemination and long-term colonization of distal tissues. We observed a reduction in Δ*flgV B. burgdorferi* expansion in the skin inoculation site, during the first week of infection by needle inoculation. This was followed by attenuation of hematogenous spread and a delay in Δ*flgV* spirochetes colonizing distal tissues. Given the complex transcriptional architecture of the *flgV* region, which affects *cis*-complementation, and the challenges of controlling FlgV induction from $P_{ind}$ in mice with IPTG added to drinking water, we examined the effect of constitutive FlgV expression from $P_{con}$. Interestingly, spirochetes overproducing the FlgV protein (++*flgV B. burgdorferi*) did not survive mouse infection and likely were eliminated at the skin inoculation site before dissemination. Thus, as we observed in culture, reduced and elevated levels of FlgV have consequences in mice.

The infection phenotypes associated with altered *flgV* levels may be explained by the innate immune response effectively eliminating less-motile spirochetes, the inability of *flgV*-mutant spirochetes to migrate through physical barriers (connective tissue, bloodstream wall, etc.), or both. Interestingly, the Δ*flgV B. burgdorferi* that do survive the initial steps in infection and have a slower kinetics of tissue colonization compared to WT spirochetes, ultimately colonize heart and joint tissue at a similar level to WT spirochetes. Yet, we observed a strong attenuation of *flgV*-mutant spirochetes for colonization of ear tissue. It was previously reported that Δ*bbO268* (*flgV*) *B. burgdorferi* are not infectious, as no Δ*bbO268* (*flgV*) spirochetes were detected from

reisolation attempts using ear punches at 3 weeks or using ear punches, joints and bladders at 5 weeks post-intraperitoneal needle inoculation of spirochetes[19]. However, infecting mice by intradermal needle injection is more physiologically similar to a natural *B. burgdorferi* infection by tick-bite, than intraperitoneal needle injection, which may account for differences between the previous infection study[19] and our data. Some studies have suggested that motility is absolutely required for *B. burgdorferi* infectivity[9,38,66]. However, our findings suggest that once spirochetes have disseminated to a tissue, motility may not be as critical for survival compared to initial skin-localized *B. burgdorferi* infection at the inoculation site.

In all, this work encompasses the first investigation of FlgV as a modulator of flagellar assembly in bacteria. We tested the consequences of defective motility during tick and mammalian infection, revealing even slight changes to the number/length of flagellar filaments have significant consequences for *B. burgdorferi* infectivity. Experiments to test when FlgV, and other proteins that modulate motility, are active during tick-mammalian infection, will be critical to define the hierarchical, flagellar regulatory network in *B. burgdorferi*. As we gain insights into how spirochetes traverse, replicate and divide inside a host, we will better understand the biology of Lyme disease.

## Methods

### Bacterial strains and plasmids

Derivatives of *E. coli* K12 MG1655 (WT) or *B. burgdorferi* B31 A3 (WT; Adams lab strain number PA003, which lacks cp9) were used for all experimental studies. The genetically tractable and infectious *B. burgdorferi* B31 A3-68Δ*bbe02* strain (WT; Adams lab strain number PA001, derivative of PA003, which lacks cp9, lp56 and gene *bbe02* on lp25[67]), was used for chromosomal mutants and plasmid transformations, as indicated. All strains, plasmids, oligonucleotides, and synthesized DNA sequences used are listed in Supplementary Data 1. Chromosomal mutants were verified by PCR and sequencing and/or whole genome sequencing. Engineered plasmid inserts were verified by sequencing. All *B. burgdorferi* transformants were verified by PCR to contain the endogenous plasmid content of the parent strain, using a panel of primers[68].

All *B. burgdorferi* chromosomal mutations were performed by allelic exchange. The *B. burgdorferi bb0268*(*flgV*)-3XFLAG construct, harboring a spectinomycin/streptomycin (P$_{flaB}$-*aadA*) resistant cassette, was PCR amplified using primers MJ1049 + MJ1050, PA102 + PA103, MJ1051 + MJ1052 with an overlap extension strategy[69]. The final PCR product was ligated with linear pCR-Blunt using a Zero Blunt PCR cloning kit (Life technologies) in DH5α *E. coli*. Twenty micrograms of pCR-Blunt-*flgV*-3XFLAG were transformed into *B. burgdorferi* B31 WT (PA003), resulting in strain PA007. Two independent *B. burgdorferi* Δ*flgV*(*bb0268*) strains were created for this study. Allelic exchange constructs were generated using primers PA337 + PA352, PA102 + PA103, PA353 + PA340 and the final PCR-generated linear DNA was either ligated into pCR-Blunt (as above) or directly transformed into WT *B. burgdorferi* (PA001), resulting in Δ*flgV* strains PA011 and PA251, respectively. The Δ*fapA*(*bb0267*) strain was created by generating an allelic exchange construct using primers PA680 + PA681, PA102 + PA103, PA682 + PA683; the final PCR-generated linear DNA was directly transformed into WT *B. burgdorferi* (PA001) resulting in strain PA395 (Δ*fapA*). The *gfp-flgV B. burgdorferi* strain was made by gene synthesis of the allelic exchange construct (*flgV*-ORF fused to *gfp*, *flaBp-aadA*, and 500 bp of the surrounding genomic region; Supplementary Data 1) and cloned into pUC57 (GenScript). A previously published *gfp* sequence[39], codon-optimized for *B. burgdorferi* was used. PCR from this plasmid (primers PA667 + PA672) was performed and 20 µg of the PCR-product was directly transformed into WT *B. burgdorferi* (PA001), resulting in strain PA402. The *B. burgdorferi* Δ*rpoS* strain, harboring a gentamycin (*flgBp-aacC1*) resistant cassette, was PCR amplified using primers PA411 + PA412, PA415 + PA416, PA413 + PA414 and the final PCR

product was directly transformed into WT *B. burgdorferi* (PA001), resulting in strain PA235.

All engineered plasmids utilized the *B. burgdorferi* shuttle vector, pBSV2G[36]. To generate an *E. coli hfq* expression vector in *B. burgdorferi*, the *E. coli* Hfq ORF was codon-optimized for *B. burgdorferi* with the OptimumGene™ algorithm and synthesized (Supplementary Data 1). The codon-optimized gene was PCR amplified with primers PA154 + PA155. These PCR products were subsequently cut with *Nde*I and *Bam*HI restriction enzymes and ligated into a pBSV2G derivate that harbors a constitutively active *flaB* promoter[70]. This resulted in p$_{con}$+*E. coli hfq*. To generate a *flgV* overexpression strain (++*flgV*), the native *B. burgdorferi flgV* ORF was PCR amplified with primers PA156 + PA157, using *B. burgdorferi* A3 genomic DNA as template. These PCR products were subsequently cut with *Nde*I and *Bam*HI restriction enzymes and ligated into a pBSV2G derivate that harbors a constitutively active *flaB* promoter[70]. This resulted in p$_{con}$++*flgV*. To create an IPTG inducible plasmid on pBSV2G (p$_{ind}$), we inserted the *flaB* promoter-*lacI* and *pQE30* promoter from pJSB252[37] onto empty pBSV2G and replaced the *flaB* promoter on p$_{con}$++*flgV* using NEBuilder HiFi DNA Assembly (New England Biolabs) to create p$_{ind}$ and p$_{ind}$+*flgV*, respectively. Briefly, high-fidelity PCR reactions (NEB Q5 DNA Polymerase) with pBSV2G and p$_{con}$++*flgV* were performed using primers PA470 + PA467 and PA466 + PA467, respectively, and the resultant PCR product gel extracted using a NucleoSpin Gel and PCR Clean-up kit (Macherey-Nagel). *flaB* promoter-*lacI* and *pQE30* were Q5 PCR-amplified from pJSB252 using primers PA468 + PA471 (for p$_{ind}$ template) and PA468 + PA469 (for p$_{con}$++*flgV* template) and mixed with the corresponding linear, gel-purified vector at a 2:1 ratio in a 20 µl NEBuilder reaction at 50 °C for 1 h. Ten microliters were subsequently transformed into NEB Turbo Competent *E. coli*.

### Growth conditions

*E. coli* strains were grown with shaking at 250 rpm at 37 °C in LB rich medium supplemented with 10 µg/ml gentamycin when applicable. Overnight *E. coli* cultures were diluted to an OD$_{600}$ of 0.05 and grown to the indicated time point.

*B. burgdorferi* were cultivated in liquid Barbour-Stoenner-Kelly (BSK) II medium supplemented with gelatin and 6% rabbit serum[71] and plated in solid BSK medium, as previously described[72]. *B. burgdorferi* cultures were grown at 35 °C with 2.5% $CO_2$ in the presence of 50 µg/ml streptomycin or 40 µg/ml gentamycin when applicable. IPTG (100 µM) was added at the subculture for clones harboring p$_{ind}$ derivates. *B. burgdorferi* starter cultures were subcultured only once from the freezer stock, by diluting spirochetes at $1 \times 10^5$ cells/ml, for all experiments.

### Growth analysis

*E. coli* growth curves were performed in biological triplicate, by measuring the OD$_{600}$ every 60 min for 780 min total, after a dilution of the overnight culture to an OD$_{600}$ of 0.05. *B. burgdorferi* growth curves were performed in biological triplicate, by diluting spirochetes to $1 \times 10^5$ cells/ml and monitoring cell growth at the indicated time points by darkfield microscopy enumeration.

### RNA-Protein coimmunoprecipitation (Co-IP) assay

Co-IP of Hfq, BB0268 (FlgV), and KhpB proteins were performed using the initial steps of the RIL-seq protocol[73]. Samples were crosslinked with 80,000 µJ/cm$^2$ of UV irradiation, lysed with 0.1 mm glass beads, and co-IPs carried out using 3 µg monoclonal anti-FLAG M2 antibody (Sigma-Aldrich) or 30 µg polyclonal FlgV antibody or 30 µg polyclonal KhpB antibody and 20 µl of Pierce protein A/G magnetic beds. Polyclonal antibodies to KhpB were generated by immunizing rabbits with purified His-tagged recombinant KhpB protein, followed by affinity purification (GenScript). Aliquots of total lysate, bead supernatant, and

IP beads were mixed with equal volume 2X Laemmli sample buffer (Bio-Rad) and subjected to immunoblot analysis as below. Aliquots of total lysate and bead supernatant were mixed with TRIzol (Thermo Fisher Scientific) and RNA extracted according to the standard protocol. RNA elution buffer (50 mM $Na_3PO_4$, 300 mM NaCl, 300 mM Imidazole, 0.1 U/µl recombinant RNase inhibitor) was added to the IP beads prior to addition of TRIzol and RNA extraction. RNA from all samples was resuspended in 12 µl of DEPC $H_2O$ and analyzed using an Agilent 4200 TapeStation System or northern blot analysis, as described below. In addition to the data included in this manuscript, we varied the amount of *B. burgdorferi* cells, growth phase and UV intensity for cross-linking, but all trials resulted in no enrichment of RNA with BB0268.

## RNA isolation

*E. coli* cells corresponding to the equivalent of 10 $OD_{600}$ or ~45 ml of a *B. burgdorferi* culture were collected by centrifugation, washed once with 1X PBS (1.54 M NaCl, 10.6 mM $KH_2PO_4$, 56.0 mM $Na_2HPO_4$, pH 7.4) and pellets snap frozen in liquid $N_2$. RNA was isolated using TRIzol (Thermo Fisher Scientific) exactly as described previously[23]. RNA was resuspended in 20–50 µl DEPC $H_2O$ and quantified using a NanoDrop (Thermo Fisher Scientific).

## Northern blot analysis

Northern blots were performed using total RNA exactly as described previously[23]. For small RNAs, 5 µg of RNA were fractionated on 8% polyacrylamide urea gels containing 6 M urea (1:4 mix of Ureagel Complete to Ureagel-8 (National Diagnostics) with 0.08% ammonium persulfate) and transferred to a Zeta-Probe GT membrane (Bio-Rad). For longer RNAs, 10 µg of RNA were fractionated on a 2% NuSieve 3:1 agarose (Lonza), 1X MOPS, 2% formaldehyde gel and transferred to a Zeta-Probe GT membrane (Bio-Rad) via capillary action overnight. For both types of blots, the RNA was crosslinked to the membranes by UV irradiation. RiboRuler High Range and Low Range RNA ladders (Thermo Fisher Scientific) were marked by UV-shadowing. Membranes were blocked in ULTRAhyb-Oligo Hybridization Buffer (Ambion) and hybridized with 5′ $^{32}$P-end labeled oligonucleotides probes (listed in Supplementary Data 1). After an overnight incubation, the membranes were rinsed with 2X SSC/0.1% SDS and 0.2X SSC/0.1% SDS prior to exposure on film. Blots were stripped by two 7-min incubations in boiling 0.2% SDS followed by two 7-min incubations in boiling water.

## Immunoblot analysis

Polyclonal antibodies to FlgV, FapA, FlaB, and RpoS were generated by immunizing rabbits with purified His-tagged recombinant FlgV protein (amino acids 58–159), recombinant FapA, recombinant FlaB, or recombinant RpoS protein followed by affinity purification (GenScript). Immunoblot analyses using *B. burgdorferi* lysates and probing for FlgV (Fig. 3A), FapA (Fig. S5C), FlaB (Fig. S8D) and RpoS (Fig. S8E) confirmed the specificity of the antibodies generated in this study. *E. coli* immunoblot analyses were performed as described previously[22], with minor changes listed below. *B. burgdorferi* total protein lysates were collected by centrifugation and cell pellets were washed twice with 1X HN (50 mM Hepes, 50 mM NaCl, pH 7.5), before boiling with Laemmli Sample Buffer + β-mercaptoethanol (Bio-Rad). Samples were separated on a Mini-PROTEAN TGX 4%–20% Tris-Glycine gel (Bio-Rad) and transferred to a nitrocellulose membrane (Thermo Fisher Scientific). Membranes were blocked in 1X TBST containing 5% milk and probed with a 1:2000 dilution of monoclonal α-FLAG-HRP (Sigma-Aldrich), 1:5000 dilution of *E. coli* α-RpoS (gift from Sue Wickner, NIH), 1:5000 dilution of α-Hfq[22], 1:500 dilution of α-FlgV (this study), 1:500 dilution of α-FapA (this study), 1:200 dilution of α-FlaB[74] or 1:20,000 dilution of α-FlaB (this study), 1:1000 dilution of α-FlgG[17,18], 1:2000 dilution of α-FliF[17], 1:500 dilution of α-RpoS (this study), 1:1000

dilution of α-OspC (Rockland), or 1:2000 dilution of α-SodA[75]. Secondary peroxidase-labeled α-mouse (GE Healthcare), α-rabbit (GE Healthcare) or α-rat (Invitrogen) HRP-conjugated antibodies were subsequently used at a 1:10,000 dilution, when required. Membranes were developed with either SuperSignal West Pico PLUS or West Femto Maximum Sensitivity Chemiluminescent Substrate (Thermo Fisher Scientific) on a Bio-Rad ChemiDoc MP Imaging System. Membranes were stripped for 15 min with Restore Immunoblot Blot Stripping Buffer (Thermo Fisher Scientific).

## Protein sequence and structure analysis

Sequence-profile searches were performed with the PSI-BLAST program against the NCBI non-redundant (nr) database clustered down to 50% sequence identity using the MMSEQS program with a profile-inclusion threshold was set at an e-value of 0.01. Iterative HMM searches were performed using the JACKHMMER program from the HMMER3 package. Profile-profile searches against PDB, Pfam, and an in-house collection of profiles maintained by the Aravind lab were performed using the HHpred program. The FAMSA and MAFFT programs were used to construct multiple sequence alignments (MSAs). TM segments and signal peptides were predicted using a deep neural network as implemented in SignalP 6 and HMM-based TMHMM[76] and Phobius programs.

Secondary structures were inferred using the JPred3 program with MSAs as inputs[77]. PDB coordinates of structures were retrieved from the Protein Data Bank. Structural models were generated using the RoseTTAfold[78], utilizing the NIH Biowulf cluster to run the GPU-dependent steps, or AlphaFold3[28] (accessed on May 29, 2024; output with the highest confidence were prepared using ChimeraX[79]). Multiple alignments of related sequences (> 30% similarity) were used to initiate HHpred searches[80] for the step of identifying templates to be used by the neural networks deployed by these programs.

Clustering of protein sequences was done using the MMSEQS program by empirically adjusting the length of aligned regions and bit-score densities[81]. Phylogenetic analysis was performed using the maximum-likelihood method with the LG or JTT models with the FastTree program[82]. The same program was used to generate distance matrices used in separating groups based on differential evolutionary rates. A custom PERL script was used to extract gene neighborhoods from genomes retrieved from the NCBI Genome database. These were then clustered using MMSEQS and filtered using a neighborhood distance cutoffs and phyletic patterns to identify conserved gene neighborhoods.

The output from the MSAs and phylogenetic analyses are published as Source Data (available at FigShare: https://doi.org/10.6084/m9.figshare.27005029).

## *B. burgdorferi* cell length analysis

*B. burgdorferi* were collected by centrifuged at 2000 x *g* for 10 min at 25 °C and washed twice in sterile 1X PBS. Cell pellets were resuspended in 1X PBS and 3 µl were immobilized on a glass slide (FisherScientific) with 18x18-1.5 cover glass (FisherScientific). *B. burgdorferi* were imaged using Zeiss Axiolab 5 microscope with a 40X/0.65 NA objective. Images were captured using an Axiocam 305 color camera (Zeiss) and cells were measured using the curve (spline) tool with ZEN 3.4 (blue edition) software.

## *B. burgdorferi* motility plate assay

Motility plate assay was adapted from[83]. On the day of the assay, a solution of 0.625% agarose was autoclaved and combined with *B. burgdorferi* plating medium for a final agarose concentration of 0.35%. 36 ml of the combined medium and agarose was added to a petri dish and left to dry in a biosafety cabinet for 1 h. A 10 ml subculture of *B. burgdorferi* was grown to logarithmic phase. Cells were centrifuged at 2000 x *g* for 20 min, resuspended in BSKII to a final

density of ~5 x $10^5$ cells/µl, and 2 µl of cell suspension were inoculated just under the agarose surface. The plates were left uncovered in a biosafety cabinet for 30 min and then incubated at 35 °C with 2.5% $CO_2$ for 9 days. The diameters of the motility rings were measured using a ruler, measuring two perpendicular diameters for each ring and calculating the mean for each sample. Measurements were made to the nearest 1 mm.

## FlgV localization

*B. burgdorferi* protein lysates were separated into soluble and membrane fractions, as performed in ref. 84. Briefly, spirochetes were grown to logarithmic phase ( ~ 2.2 x $10^7$ cells/ml) in BSKII medium, spun at 3210 x *g* for 10 min, washed twice with cold 1X HN buffer, and resuspended in 1 ml 1X HN with 100 µl Halt Protease Inhibitor Cocktail (Thermo Fisher Scientific). Spirochetes were lysed by sonication and spun at 125,000 x *g* to collect the soluble and membrane components of the lysate.

## Fluorescence microscopy

*B. burgdorferi* were centrifuged at 2000 x *g* for 10 min at 25 °C and resuspended in sterile 1X PBS at a final density of ~1 × $10^5$ cells/µl. 3 µl of cells were immobilized on a 35 mm glass-bottom dish (MatTek) with a 1 cm disc of 0.77% agarose. *B. burgdorferi* were imaged using Zeiss LSM800 confocal microscope with a 63X/1.4 NA oil objective. Green fluorescence and bright-field images were captured for each field-of-view. Images were processed with software Fiji ImageJ (version 2.1.0/1.53c).

For demography analysis, images were analyzed with the software Fiji ImageJ2 (version 2.3.0/1.53q). The fluorescence intensity was plotted free-hand along each cell's length using the Plot Profile tool on the fluorescent channel, yielding a data frame for each cell. A total of 300 cells were analyzed for each growth condition. Using R (version 4.2.1) and RStudio (version 2022.07.0, build 548), the data frames were organized and depicted in a demograph using the tidyverse[85], dplyr[86], and ggplot2 packages[87].

## Cryo-ET sample preparation, data collection, and reconstruction

*B. burgdorferi* were grown to a density of: 3.5 x $10^7$ cells/ml for WT/p (PA023), 1.05 x $10^7$ cells/ml for WT/$p_{con}$ + +*flgV* (PA267), 4.5 x $10^6$ cells/ml for WT/$p_{ind}$ (PA273), 7.5 x $10^5$ cells/ml for Δ*flgV*/$p_{ind}$ (PA310), 6.0 x $10^6$ cells/ml for Δ*flgV*/$p_{ind}$+*flgV* (PA312), and 2.0 x $10^7$ cells/ml WT *flgV-gfp* (PA402). Cultures were shipped overnight to the Microbial Sciences Institute at Yale University, where they were immediately processed. Cultures were centrifuged at 5,000 rpm and washed three times with 1X PBS. 10 µl of resuspended *B. burgdorferi* (normalized to an $OD_{600}$ = 1.0) were mixed with 20 µl 10-nm colloidal gold particles. The mixture was deposited onto freshly glow-discharged grids (R2/1 Quantifoil) for 1 min, blotted with filter paper for 3 seconds, and frozen in liquid ethane using a gravity-driven homemade plunger apparatus, as described previously[40].

Bacterial samples were imaged with a Titan Krios microscope (Thermo Fisher Scientific) equipped with a field emission gun, an energy filter, and a K3 direct-detection device (Gatan). An energy filter with a slit width of 20 eV was used for data acquisition at an ~6 µm defocus.

To determine the flagellar basal body structures, we only collected tilt series from the cell poles at a magnification of x42,000, resulting in a pixel size of 2.148 Å. For each tilt series, we collected 33 image stacks at a range of tilt angles of between +48° and −48° (3° step size) using a bidirectional scheme with a cumulative dose of 60e⁻/Å². All the title series were aligned and reconstructed by IMOD[88,89]. In total, we generated 120 tomograms from WT/p (PA023), 64 from Δ*flgV*/$p_{ind}$ (PA310), 41 from Δ*flgV*/$p_{ind}$+*flgV* (PA312), 190 from WT *flgV-gfp* (PA402), and 65 from WT/$p_{con}$ + +*flgV* (PA267) *B. burgdorferi*.

To visualize flagella and cell poles for over 2 µm length, multiple tilt series were collected along the spirochetes by SerialEM[90]. For each tilt series, we collected 33 image stacks at a range of tilt angles of between +48° and −48° (3° step size) using a bidirectional scheme with a cumulative dose of 80e⁻/Å² at a magnification of x19,000, resulting in a pixel size of 4.556 Å at the specimen level. All the title series were aligned and reconstructed by IMOD[88,89]. In total, we generated 21 tomograms from WT/$p_{ind}$ (PA273), 21 from Δ*flgV*/$p_{ind}$ (PA310), 18 from Δ*flgV*/$p_{ind}$+*flgV* (PA312), 18 tomograms from WT/p (PA023), and 18 from WT/$p_{con}$ + +*flgV* (PA267) *B. burgdorferi*. Flagellar basal bodies and flagellar filaments were manually tallied for each tomogram.

## Cryo-ET data visualization

Representative reconstructions from *B. burgdorferi* strains PA023, PA310, PA312 and PA267 and were visualized in IMOD[88]. Periplasmic flagella, the basal body, the outer membranes, and inner membranes were manually segmented in IMOD.

## Subtomogram averaging

To determine flagellar basal body structures by subtomogram averaging, we visually identified the basal bodies from the tomograms. In total, we found 907 basal bodies from PA023, 562 from PA310, 272 from PA312, 1462 from PA402, and 485 from PA267. I3 package[91] was used to determine in situ structures as described previously[40].

## Ethics statement

The National Institutes of Health and the University of Central Florida are accredited by the International Association for Assessment and Accreditation of Laboratory Animal Care. All animals were cared for in compliance with the Guide for the Care and Use of Laboratory Animals. Protocols for all animal experiments were prepared according to the guidelines of the National Institutes of Health, reviewed and approved by the *Eunice Kennedy Shriver* National Institute of Child Health and Human Development Institutional Animal Care and Use Committee or the UCF Institutional Animal Care and Use Committee.

## Infection of *Ixodes* larvae by immersion and tick feeding

Approximately 2-month-old *Ixodes scapularis* naïve larval ticks (Oklahoma State University, Department of Entomology and Plant Pathology) were infected by immersion[46]. Larval ticks were dehydrated by exposure to saturated ammonium sulfate for ~88 h. *B. burgdorferi* strains were grown to logarithmic phase (2 x $10^7$ - 5.3 x $10^7$ cells/ml) and diluted to 2 x $10^7$ cells/ml in BSKII. 500 µl of spirochetes were incubated with dehydrated ticks at 35 °C for 1.5 h and washed twice with sterile 1X PBS. The inoculum cultures were verified to contain the expected endogenous plasmids, using a panel of primers[68]. Individuals from each inoculum clone were analyzed for the presence of virulence plasmids lp25, lp28-1 and lp36[92], each of which was confirmed to be present in 80–100% of the population.

Twelve days following the artificial infection, groups of 10 unfed larvae were surface sterilized and plated in solid BSKII containing RPA cocktail (60 µM rifampicin, 110 µM phosphomycin, 2.7 µM amphotericin B) and 40 µg/ml gentamycin to determine *B. burgdorferi* CFUs/10 larvae. Two days following the artificial infection, groups of ~129 artificially infected larvae, on average, were fed to repletion on groups of 3 naïve, 6–8-week-old, C3H/HeN mice (Envigo). Mice were assayed for *B. burgdorferi* infectivity by reisolation in BSKII, as described in the section below. Ticks fed for ~1 week before naturally falling off. Approximately 1 week following the tick collection, individual fed larvae were analyzed for *B. burgdorferi* burden by plating, as described above. A subset of fed larvae were allowed to molt into nymphs, and ~5 weeks following, individual unfed nymphs were analyzed for *B. burgdorferi* burden by plating, as described above. Groups of ~5 unfed nymphs, on average, were fed on groups of 3 naïve, 6–8-week-old, C3H/HeN mice (Envigo) to repletion. Ticks fed for ~1 week before naturally falling off.

Approximately 1 week following the tick collection, individual fed nymphs were analyzed for *B. burgdorferi* burden by plating, as above.

## Mouse infection by needle inoculation

All *B. burgdorferi* strains were grown to stationary phase (~2-3 × 10$^8$ cells/ml). Groups of 6 C3H/HeN mice (Envigo) each were needle-inoculated with 1 x 10$^4$ *B. burgdorferi* intradermally. Inoculum densities were confirmed by plating for individuals in a solid medium. All inoculum cultures were verified by PCR to contain the expected endogenous plasmids[68,93]. Individuals from each inoculum clone were analyzed for the presence of virulence plasmids lp25, lp28-1 and lp36[92], each of which was confirmed to be present in 90–100% of the population.

## Determination of *B. burgdorferi* density in blood

On day 6 post-inoculation approximately 50 μl of blood was collected from each mouse into K2 EDTA MiniCollect blood collection system tubes (Greiner). Each 50 μl whole blood sample was added to 1 ml BSKII and then plated in solid BSK-agarose medium plus RPA cocktail (60 μM rifampicin, 110 μM phosphomycin, and 2.7 μM amphotericin B) and plates incubated at 35 °C under 2.5% $CO_2$ for 7–14 days. Colony-forming units (CFU) were enumerated and *B. burgdorferi*/ml of blood was calculated as the CFU/volume of blood analyzed multiplied by 100 for each sample[50].

## Spirochete reisolation and quantification of *B. burgdorferi* densities in tissues

Mouse infection was determined by cutting each mouse tissue in half and assaying spirochete reisolation and quantitative PCR for *B. burgdorferi* loads for skin inoculation site, ear, heart, and joint samples (exactly as described previously[50,93]). TaqMan probe and primers (IDT DNA) specific to the *B. burgdorferi flaB* gene (primers: MJ1137, MJ1138, MJ1139) and mouse *nid* gene (primers: MJ1140, MJ1141, MJ1142) (Supplementary Data 1) were used to measure their respective copy numbers by qPCR and a standard curve approach, using 100 ng of DNA extracted from mouse tissues as the template.

## Reporting summary

Further information on research design is available in the Nature Portfolio Reporting Summary linked to this article.

# Data availability

Source data are provided with this paper. All uncropped and unprocessed images with molecular weight markers labeled are available on Figshare (https://doi.org/10.6084/m9.figshare.27005029). Source data are provided with this paper.

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

## Acknowledgements

Thank you to the late C. Savage, a tenacious and inspirational Borreliologist and colleague, for pointing out that *bb0268* is encoded near flagellar genes. Thank you to J. Botting for indicating to us that BB0268 is a homolog to FlgV in *H. pylori*. Thank you to C. Li, S. Gottesman, S. Melamed and members of the Adams and Storz lab for helpful comments on the manuscript. Thank you to A. Zhong for assistance with the AlphaFold 3 analysis. We also thank T. Li and J. Iben for whole genome sequencing and analysis. Thank you to J. Seshu for providing the α-SodA antibodies, C. Li for the α-FlgG and α-FliF antibodies and M. Motaleb for the ΔflaB and ΔflaB+flaB *B. burgdorferi*. This work utilized the computational resources of the NIH HPC Biowulf cluster (https://hpc.nih.gov). This work was supported by the National Institutes of Health Independent Research Scholar Program [P.P.A.]; the *Eunice Kennedy Shriver* National Institute of Child Health and Human Development Intramural Research Program [1ZIAHD008995 to P.P.A.] and [1ZIAHD01608 to G.S.]; a National Institute of General Medical Sciences Postdoctoral Research Associate (PRAT) fellowship [1Fi2GM133345 to P.P.A]; the National Institute of Allergy and Infectious Diseases [R01AI099094 to M.W.J.], [R01AI087946 to J.L.], [R01AI132818 to J.L.]; the National Library of Medicine [1ZIALM594244 to L.A.]; the National Research Fund for Tick-Borne Diseases [M.W.J.]; and the Yale CryoEM Resource [1S10OD023603].

## Author contributions

P.P.A., G.S., and M.W.J. conceived the project. P.P.A., G.S., M.W.J., J.L, and L.A. funded the project. P.P.A., G.S., M.W.J., J.L, and L.A. designed the experiments. M.Z.C., D.S., P.P.A., H.Y., A.G.L., Y.Y.C, J.L.S., and L.A. performed the experiments. P.P.A., G.S., M.W.J., L.A., and J.L. wrote the manuscript, with contributions from all authors. All authors critiqued and edited the final manuscript.

## Funding

## Competing interests

The authors declare no competing interests.
