## [Transparent Peer Review file · Nature Communications]

Broadly conserved FlgV controls flagellar assembly and *Borrelia burgdorferi* dissemination in mice

Corresponding Author: Dr Philip Adams

Version 0:

Reviewer comments:

Reviewer #1

(Remarks to the Author)

This manuscript investigates the function of *Borrelia burgdorferi* protein BB0268, which was previously suggested to be a homolog to the RNA-binding protein Hfq in *E. coli*. By contrast, the authors find no evidence for BB0268 interacting with RNA. Instead, the present study presents evidence for BB0268 being a flagellar protein, which they rename FlgV. The authors show that both deletion and overexpression of FlgV results in impaired cell division and motility in *B. burgdorferi*, while complementation of FlgV at physiological levels phenocopies a wild-type strain. Cell fractionation and cryo-ET indicate that FlgV is an integral part of the flagellar motor complex and affects the assembly of flagellar filaments. Finally, the authors show that both deletion and overproduction of FlgV affects dissemination of *B. burgdorferi* in mice.

The manuscript is very well written and the data is generally of very high quality. The findings of FlgV as a conserved flagellar protein is interesting and convincing. The data presented in the first part of the manuscript convincingly argues for that the previous suggestion that FlgV is an Hfq homolog most likely is wrong.

Major comments:

1. The interpretation of the Northern blot analysis of flgV mRNA(s) was not clear to me (Fig. S5 and associated manuscript text). First, the replacement of the 480 nt flgV ORF by the 1160 nt resistance cassette should add 680 nt to any transcript including flgV. However, none of the bands detected with probes targeted against flanking ORFs fapA and flhG show increased size in the flgV deletion strain. The expected increased size should be especially evident for the bands between 1000 and 3000 nts, but this is not the case. The authors need to acknowledge this in the manuscript and explain. Second, based on the transcriptome mapping published previously by the authors, it should be possible to deduce the exact boundaries of the most abundant transcripts, e.g. which ORFs are included in the different transcripts. Please include this information or, if the transcriptome data does not permit such analysis, explain why not.

2. Although the present study convincingly shows that FlgV is not an Hfq homolog, the authors should mention the previous study also for the results that are consistent between the two studies. For instance, the increased cell length of the flgV deletion strain was already shown in the previous study, which should be mentioned at page 13 where these results are presented. Likewise, the previous study showed that deletion of flgV resulted in strongly decreased *B. burgdorferi* infection in mice, while complementation of FlgV in trans restored wild-type infectivity. This should be clearly stated in the Results section (page 18-22). As these parts are written, it appears as if FlgV's effect on cell length and infectivity are new findings, which is not correct.

3. It was unclear why the authors chose to not include the Pind+flgV strain for complementation during the infection experiments. This strain nicely complements the changes observed in cell length, motility and cryoET density. Instead, the overexpression strain, which phenocopies the deletion strain, was used. If the authors have infection data with the proper complementation strain, this should be included. As of now, it is not unequivocally clear that the absence of flgV, rather than an indirect cause (e.g. polar effects in the deletion strain), is responsible for the infection phenotypes.

Minor comments:

1. Figure 1A: It was surprising that the FlgV appears to be much more abundant than Hfq. Hfq is known to be a highly abundant protein in *E. coli*. Still, FlgV is at least ten times more abundant according to the Western blot. Did the authors use the same number of cells as input in the colP experiments?
2. Figure 1 B-D: The colP results convincingly shows that Hfq and KhpB pulls down considerably more RNA than FlgV. It was, however, surprising that the authors did not include quantification of RNA from a mock-IP, e.g. FLAG-mediated IP with a non-tagged strain, since that would give a baseline value for unspecific RNA pull-down. If the authors have this information, it would be advisable to include this in Fig. 1.
3. Consider shortening the text related to Figure 1, and move Fig. 1 to Supplemental information. In my opinion, the strength of the manuscript lies in the characterization of FlgV's function, which is interesting and convincing. Although important, the description of results proving what FlgV is not could be considerably shortened.

Reviewer #2

(Remarks to the Author)

The manuscript by Zamba-Campero et al., characterizes a previously mischaracterized protein, now called FlgV, in *Borrelia burgdorferi* and show that genetically, cell biologically and structurally associates with the flagellum in which both overexpression and absence impair flagellar filament assembly. The data are gorgeous and the toolkit is broad. The paper involves protein purification, pulldowns, RNA interaction assays, alphafold modeling, phylogenetic analysis, subcellular localization, quantitative microscopy, EM tomography, fundamental genetic analysis, physiological phenotyping, antibody generation, all before getting into infection models of both the tick and mouse hosts. The work is a technical tour-de-force told with writing that is exceptionally clear. The paper unfortunately feels too long, and tells three somewhat separate stories, which while internally convincing don't seem to synergize.

This first story deals with proving FlgV isn't Hfq. The work is excellent (albeit mostly in supplemental) but dwells in demonstrating another paper's conclusions are wrong from multiple angles. The first story feels like it could be a separate manuscript (breaking after the phylogenetic argument), or cut to a paragraph or two needed to effectively make the point necessary to get to the next story: if FlgV isn't Hfq, what is it? This deserves publication because it is so well done and comprehensive but seems more of a distraction in this context and packaging it all in supplemental felt almost tragic.

The second story deals with the molecular analysis of FlgV. The authors convincingly demonstrate that it is a conserved part of the flagellar regulon and that the protein interacts with the flagellar basal body. I think the flgV phenotype is a partial down (rather than out) for motility and the motility assays would benefit from including a fully non-motile mutant for comparison. If true, the phenotype is rather subtle, and the mechanism of promoting filament completion on a subset of flagella is unclear. I note that in terms of storytelling the authors could have chosen to study FlgV by reverse genetics based on the conserved but unknown function of the protein as easily as choosing to study FlgV by reverse genetics based on its reported role as Hfq without altering the major outcomes of the analysis.

The third story deals with the role of FlgV in the *Borrelia* infection cycle. Skillfully using what appear to be remarkably well-developed tick and mouse model systems (perhaps new (?) or perhaps from the literature), the authors show that FlgV is needed for dissemination and propagation in the mouse but not the tick. The data are beautiful and the interpretations clean and careful. These results could be novel but I'd be surprised if motility defects haven't already been shown to cause a defect in mammalian infection. This story and the detailed dissection of the infection cycle seems like it could also be a stand alone work, and would also benefit from having a direct comparison to a fully non-motile mutant. It would be useful to know whether flgV phenocopies a full flagellar defect or manifests as an intermediate phenotype.

To be clear, I think the paper could be published as is, but I feel that the excessive length and somewhat tenuous links between stories causes nuanced points to get lost and almost draws attention to weaknesses. I note that the intro contains a seemingly extraneous paragraph on FlhF/FlhG and the discussion could be trimmed down as well, since as near as I can tell, there isn't much mechanistic clarity to explain the partial filament phenotype. If I had to guess, the FlgV mutant might have a reduced rate of type III secretion (but I imagine this would be quite challenging to test in a spirochete and I'm not suggesting it is necessary for publication).

Minor comments:

Line numbers would aid review.

Abstract: "Flagellar motors" is a bit of a loaded term in the field and could suggest that FlgV interacts with either the stators or the C-ring to control power. But the point of the paper I believe is that FlgV controls flagellar assembly and confusion could arise since the EM data places FlgV near the rotor. Consider changing to "flagella" or "flagellar basal bodies".

p. 10 top. I do not think that the promoter of the flgB operon is consensus for s70. Double check. The -10 looks to be consensus but the -35 is not.

p.12. "flagellar filamentation" is potentially problematic as filamentation is often a word to describe cell morphology (and an elongated or filamentous cell phenotype appears later in the manuscript). Consider "flagellar filament synthesis" or "expression of a regulon that includes the flagellar filament".

p. 26. Please clarify what is meant by “intramembrane flagellar sensor”. I think it is an important idea but not quite sure what the authors are thinking in terms of FlgV.

Reviewer #3

(Remarks to the Author)

Borrelia burgdorferi (Bb) is a spirochetal bacterium that causes Lyme disease. Motility via their endoflagella is essential for migration between tick and vertebrate hosts via dense tissues, thus there is interest in understanding the mechanisms that control this essential virulence function. The current study was initially intended to further identify and assess the RNAs bound by the previously reported putative “Hfq-like” BB0268 protein. However, the authors initial studies using multiple well-controlled immunoprecipitation methods were not able to detect any significant RNA-binding properties, suggesting a mischaracterization. Multiple in silico searches for Hfq homologs in Bb provided no significant similarities, though other categories of RNA binding proteins were present. Alpha Fold predictions for BB0268 possessed no similarities to Hfq proteins and subsequent attempts to recapitulate the findings of the previous publication identifying Bb0268 as a Hfq-like proteins were unsuccessful. A subsequent genomic survey indicated that BB0268 is situated in the flgB operon together with many other motility-related genes, and that homologs exist in many other bacterial genera, with the most closely related being flaV in *Campylobacter*. This suggested that BB0268 (FlgV) is involved in motility, and subsequent studies determined that FlgV impacts cell division and motility, that it is located in the outer membrane and is associated with the flagellar motors at the poles, that certain FlaV levels are required for assembly of flagellar filaments, and that deficiencies in FlaV has significant effects on infectivity/persistence within mouse tissues. Overall, this is a very well organized and controlled set of studies, and the manuscript is very well written to describe the significance of the experimental findings. The data provided to disprove that BB0268 is a Hfq-like protein is convincing, as is the newly proposed role in flagellar organization and motility. While there appears to be a number of significant findings, including the counterpoint to the previous BB0268 characterization, there are a few issues that should be addressed so the reader can better understand/assess the findings.

Issues

1. Introduction. The use of passive voice in the introduction leads to some slightly bulky phrasing (e.g. page 3 line 15 “Larval ticks are hatched uninfected” reads more cleanly as “Larval ticks hatch uninfected”).
2. Page 4, line 2. The phrase “overcome host immune responses” is a bit unclear and suggestive. Can this be rephrased to better reflect the authors intent.
3. Page 11, line 13. A reference should be provided regarding the statement that the listed motility-related genes may control flagellar count.
4. Page 13 and description of Figure 3C. Is the decreased growth of the flaV mutant significantly less than the WT? Should be able to perform statistics on the multiple growth curves generated to produce the graph.
5. Figure 5K. The authors state that there are differences in the intensity of certain bands, which is not really apparent by eye and there is no quantification. It would be prudent to perform densitometry on the multiple gels represented by this figure and acquire hard units to allow statistics.
6. Page 25, line 13. The authors state that “Flagella-mediated motility is likely required to efficiently separate dividing spirochetes” but provide no reference. Adding references PMID 25690096 and 25690096 may help support this statement.
7. Page 28, line 10. The authors state that “This suggests that motility is not critical for *B. burgdorferi* inside ticks”. This appears to be misleading as the mutants do maintain some motility, as evidenced by the agar diffusion assay and the dissemination in the murine host. This statement should be altered for clarity.

Reviewer #4

(Remarks to the Author)

Reviewer #5

(Remarks to the Author)

Bacteria flagellar assembly and flagella-mediated cell mobility are critical for microorganisms’ survival and pathogenesis. In

this study, Philip P. Adams and co-authors combined multiple approaches to thoroughly investigate the *B.burgdorferi* gene bb0268/flgV in flagellar regulation and assembly, and subsequently characterized its role in *B.burgdorferi* dissemination in mice. Based on these studies, the authors have reannotated the function of the flgV gene.

They first showed that bb0268/flgV is not an RNA binding protein, not an Hfq homolog that involves in flagellar regulon regulation, as claimed by a previous study, but rather a membrane-associated protein. They then demonstrated that flgV is co-expressed with flagellar motor/basal body and localized the position of flgV by using cryo-ET in the context of different isogenic strains. They showed that flgV localizes to the flagellar motor, likely between the C-ring and MS rings. The authors further demonstrated that flgV protein, either by deletion or overproduction, doesn't change the *B.burgdorferi* motor assembly, numbers, and positions, but instead affects the assembly/length of the flagellar filament, which ultimately damages the *B.burgdorferi* mobility, cell division and dissemination in mice.

Overall, the experiments are well designed, the results are convincing, and the manuscript is well-written and easy to follow. Considering the clarification of the role of flgV in the regulation of the pathogenetic bacteria *B.burgdorferi* flagellar filament and the substantial amount of work, this manuscript is suitable for publishing in Nature communications, in my opinion, and would be of great interest to researchers in the flagellar field, specifically those studying *B.burgdorferi* mobility and its mediated Lyme disease. This work would pave the way for further study to elucidate the exact mechanism of flgV in the regulation of flagellar filament length. I have only one major concern and several minor points.

My major concern is about the interpretation of the cryo-ET map. As flgV contains two transmembrane helices and has a short intracellular disordered loop region, it would be a challenge to identify its precise position from the low-resolution cryo-ET map. I have no doubt that flgV is part of flagellar motor as shown by the authors using the light microscope. I also appreciate that the authors mentioning a parallel study: "FlgV forms a flagellar motor ring that is required for optimal motility of *Helicobacter pylori*," showing a similar position of CjflgV in the motor. It seems from Figure 4C right panel that flgV mediates the interaction of the C-ring and M-ring. It would be great for the authors to check and clarify whether the density is really from flgV. Or deletion of flgV changed some part of the flagellar motor, and consequently alters the density observed by the authors. Could the authors provide some evidence of direct or indirect interaction of flgV with the flagellar MS ring protein FliF through e.g., co-IP and AlphaFold2 prediction.

Minor points

In Fig.2, the authors classify flgV genes into three types through analyzing their genome context. The author should also build a phylogenetic tree, at least using the sequences in figure S6A, to analyze the evolution of flgV genes from different bacteria species.

In Fig. 4C, the flagellar stator units need to be annotated.

In Fig. S3, 'AlphaFold2 secondary structure predictions' need to be corrected as 'AlphaFold2 tertiary structure predictions'. AF2 predicts protein 3-D dimensional folding. The authors need to correct this in the main text on page 8, in three positions. The author should also show in Fig. S3, the AF2 predicted dimer of flgV, and map the position of the conserved Arg highlighted in Fig. S6A, to infer if the Arg is critical for flgV dimerization, since the author has observed in the denatured gel that some flgV exists as a dimer.

Uniform word spelling in the text needs to be followed, e.g. either Spirochetes or Spirochaetes. In some position FlgV should be written as *Italic*, and BB0628 should be written as bb0628. Please check out throughout the manuscript.

Version 1:

Reviewer comments:

Reviewer #1

(Remarks to the Author)

The authors have clarified all issues raised in my review. I delightedly endorse the manuscript for publication.

Reviewer #2

(Remarks to the Author)

The authors have addressed my concerns.

Reviewer #3

(Remarks to the Author)

[No comments for authors]

Reviewer #4

(Remarks to the Author)

Reviewer #5

(Remarks to the Author)

The authors have addressed all my major concerns. I have only one minor question remaining: The authors mentioned that "FliF and FliA are not present in *Borrelia* sp." If so, what protein forms the MS ring in *Borrelia* sp.? FliF? Once this is clarified, I support the publication of the paper.

RESPONSE TO REVIEWER COMMENTS

Reviewer #1 (Remarks to the Author):

This manuscript investigates the function of *Borrelia burgdorferi* protein BB0268, which was previously suggested to be a homolog to the RNA-binding protein Hfq in *E. coli*. By contrast, the authors find no evidence for BB0268 interacting with RNA. Instead, the present study presents evidence for BB0268 being a flagellar protein, which they rename FlgV. The authors show that both deletion and overexpression of FlgV results in impaired cell division and motility in *B. burgdorferi*, while complementation of FlgV at physiological levels phenocopies a wild-type strain. Cell fractionation and cryo-ET indicate that FlgV is an integral part of the flagellar motor complex and affects the assembly of flagellar filaments. Finally, the authors show that both deletion and overproduction of FlgV affects dissemination of *B. burgdorferi* in mice.

The manuscript is very well written and the data is generally of very high quality. The findings of FlgV as a conserved flagellar protein is interesting and convincing. The data presented in the first part of the manuscript convincingly argues for that the previous suggestion that FlgV is an Hfq homolog most likely is wrong.

We appreciate these positive comments.

Major comments:

1. The interpretation of the Northern blot analysis of *flgV* mRNA(s) was not clear to me (Fig. S5 and associated manuscript text). First, the replacement of the 480 nt *flgV* ORF by the 1160 nt resistance cassette should add 680 nt to any transcript including *flgV*. However, none of the bands detected with probes targeted against flanking ORFs *fapA* and *flhG* show increased size in the *flgV* deletion strain. The expected increased size should be especially evident for the bands between 1000 and 3000 nts, but this is not the case. The authors need to acknowledge this in the manuscript and explain. Second, based on the transcriptome mapping published previously by the authors, it should be possible to deduce the exact boundaries of the most abundant transcripts, e.g. which ORFs are included in the different transcripts. Please include this information or, if the transcriptome data does not permit such analysis, explain why not.

It has been a challenge to completely resolve the transcriptional units/RNAs of this operon. Using a probe internal to the *bb0269* ORF, we did detect a new band (~1000 nt) by northern analysis in the Δ *bb0268* samples which could correspond to P_{flaB} -*aadA*. The resolution at 4,000-6,000 nt on the northern blot makes it difficult to detect a 680 nt change for the full operon transcript.

We performed additional northern analysis using an oligonucleotide probe complementary to the *aadA* ORF (Fig. S5B). The results of this experiment showed abundant bands \leq 2,000 nt, suggesting that the transcript encoding the *aadA* resistance cassette (1160 bp) is largely terminated or cleaved. However, we observed the ~6,000 nt band also detected with the *bb0267*, *bb0268*, *bb0269* and *bb0272* probes using the *aadA* ORF probe indicating that P_{flaB} -*aadA* is co-transcribed with *bb0272*–*bb0264* in some RNAs.

The 5'- and 3'RNA-seq approaches, published previously by our lab, indeed are excellent datasets to estimate possible transcript boundaries. However, when there are abundant 5' and

3' ends, it is difficult to delineate all the transcriptional units/RNAs, even with northern analysis using multiple probes. It seems the transcripts from this genomic region are highly processed.

We have added a statement about the complexity of the region and the difficulty in delineating all the transcripts (page 10).

2. Although the present study convincingly shows that FlgV is not an Hfq homolog, the authors should mention the previous study also for the results that are consistent between the two studies. For instance, the increased cell length of the flgV deletion strain was already shown in the previous study, which should be mentioned at page 13 where these results are presented. Likewise, the previous study showed that deletion of flgV resulted in strongly decreased *B. burgdorferi* infection in mice, while complementation of FlgV in trans restored wild-type infectivity. This should be clearly stated in the Results section (page 18-22). As these parts are written, it appears as if FlgV's effect on cell length and infectivity are new findings, which is not correct.

Where appropriate throughout the results and discussion, we have tried to report the observations that are consistent between our work and those of Lybecker et al., 2010. For example on page 14 we state: "Previous gel filtration chromatography reported that FlgV likely forms a dimer *in vitro* (Lybecker et al., 2010)" consistent with this, we observed a higher molecular weight FlgV band (Fig. 4A). Lybecker et al., 2010 reported the mutant strain "was significantly longer than wild type". We now acknowledge this report, page 13. However, we had novel observations, such as the presence of septa in "long cells" and that all cells are the same size in stationary phase.

Our infectivity data differs from what Lybecker et al., 2010 reported. Lybecker et al. performed an intraperitoneal needle injection and looked for reisolation of $\Delta bb0268$ (*flgV*) spirochetes from mouse ear punches, joints and bladders at 5 weeks post-injection. They did not reisolate any $\Delta flgV$ spirochetes from any mouse tissues. We performed a tick feeding and intradermal needle injection with $\Delta flgV$ spirochetes; both independent methods resulted in >60% of mice infected with $\Delta flgV$ spirochetes. We mention this discrepancy in the discussion: "It was previously reported that $\Delta bb0268$ (*flgV*) *B. burgdorferi* are not infectious, as no $\Delta bb0268$ (*flgV*) spirochetes were detected from reisolation attempts using ear punches at 3 weeks or using ear punches, joints and bladders at 5 weeks post-intraperitoneal needle inoculation of spirochetes. However, infecting mice by intradermal needle injection is more similar to a natural *B. burgdorferi* infection by tick-bite, than intraperitoneal needle injection, which may account for differences between the previous infection study and our data." (page 29)

3. It was unclear why the authors chose to not include the $P_{ind}+flgV$ strain for complementation during the infection experiments. This strain nicely complements the changes observed in cell length, motility and cryoET density. Instead, the overexpression strain, which phenocopies the deletion strain, was used. If the authors have infection data with the proper complementation strain, this should be included. As of now, it is not unequivocally clear that the absence of *flgV*, rather than an indirect cause (e.g. polar effects in the deletion strain), is responsible for the infection phenotypes.

We completely agree with the reviewer that genetic complementation is important. As the reviewer notes the $P_{ind}+flgV$ strain was included for complementation in all of the *in vitro* experiments presented in the manuscript. These experiments required tightly controlled IPTG-induction of *flgV* expression for robust complementation, as too much or too little *flgV* resulted in significant attenuation of *B. burgdorferi* growth. These experiments demonstrated that the

phenotypes of the *flgV* mutant were solely attributable to the absence of *flgV*. For infection experiments, IPTG must be delivered to mice through the drinking water. Thus, the amount of IPTG that the mice take in cannot be precisely controlled. Because complementation using the $P_{ind}+flgV$ strain required precise amounts of IPTG we felt that the infection results obtained from this strain may not truly reflect whether complementation was achieved *in vivo* or not. Therefore, we elected not to include the $P_{ind}+flgV$ strain in the infection studies. We now state this limitation in the discussion (pages 28-29).

We did attempt to create a *cis*-complementation of *flgV* on the chromosome at its native locus to use for mouse infection experiments. However, while this strain restored FlgV to WT levels, it significantly lowered the levels of FapA (encoded directly downstream to *flgV*):

To directly test for polar effects in the deletion, we performed northern analysis of the genes surrounding *flgV* (Fig. 5SB) and documented that transcript levels of the complex operon remain largely unchanged. We did observe that some transcripts detected with the *fapA* (*bb0267*) probe increased in the *bb0268* deletion samples. Most importantly, we generated native antibodies to FapA (Fig. S5C and above) and performed immunoblot analysis to measure FapA levels in WT and $\Delta flgV$ lysates (Fig. S5D). This showed that BB0267 levels are not altered in the $\Delta flgV$ strain (statement added to text; page 10), indicating the $\Delta flgV$ does not have polar effects.

Minor comments:

1. Figure 1A: It was surprising that the FlgV appears to be much more abundant than Hfq. Hfq is known to be a highly abundant protein in *E. coli*. Still, FlgV is at least ten times more abundant according to the Western blot. Did the authors use the same number of cells as input in the colP experiments?

We cannot directly compare the levels of FlgV in *B. burgdorferi* to Hfq in *E. coli* given the very different nature of these cells resulting in different total protein inputs as reflected in the differences in the Ponceau S staining between the *B. burgdorferi* and *E. coli* samples.

2. Figure 1 B-D: The colP results convincingly shows that Hfq and KhpB pulls down considerably more RNA than FlgV. It was, however, surprising that the authors did not include

quantification of RNA from a mock-IP, e.g. FLAG-mediated IP with a non-tagged strain, since that would give a baseline value for unspecific RNA pull-down. If the authors have this information, it would be advisable to include this in Fig. 1.

As suggested, we have added these data to Fig. 1.

3. Consider shortening the text related to Figure 1, and move Fig. 1 to Supplemental information. In my opinion, the strength of the manuscript lies in the characterization of FlgV's function, which is interesting and convincing. Although important, the description of results proving what FlgV is not could be considerably shortened.

The incorrect notion that *bb0268* encodes a homolog to *E. coli* Hfq has been perpetuated in numerous *B. burgdorferi* reviews and NCBI annotations. Additionally, other *B. burgdorferi* proteins have been suggested to be "RNA-binding proteins" without *in vivo* data or properly controlled *in vitro* experiments. Therefore, we think it is important to include Figure 1 as a main figure and present reliable approaches to test RNA association with a protein of interest in *B. burgdorferi*. We have tried to succinctly describe the experiments in Figure 1.

Reviewer #2 (Remarks to the Author):

The manuscript by Zamba-Campero et al., characterizes a previously mischaracterized protein, now called FlgV, in *Borrelia burgdorferi* and show that genetically, cell biologically and structurally associates with the flagellum in which both overexpression and absence impair flagellar filament assembly. The data are gorgeous and the toolkit is broad. The paper involves protein purification, pulldowns, RNA interaction assays, alphafold modeling, phylogenetic analysis, subcellular localization, quantitative microscopy, EM tomography, fundamental genetic analysis, physiological phenotyping, antibody generation, all before getting into infection models of both the tick and mouse hosts. The work is a technical tour-de-force told with writing that is exceptionally clear. The paper unfortunately feels too long, and tells three somewhat separate stories, which while internally convincing don't seem to synergize.

This first story deals with proving FlgV isn't Hfq. The work is excellent (albeit mostly in supplemental) but dwells in demonstrating another paper's conclusions are wrong from multiple angles. The first story feels like it could be a separate manuscript (breaking after the phylogenetic argument), or cut to a paragraph or two needed to effectively make the point necessary to get to the next story: if FlgV isn't Hfq, what is it? This deserves publication because it is so well done and comprehensive but seems more of a distraction in this context and packaging it all in supplemental felt almost tragic.

Thank you for these positive comments regarding our work to comprehensively document that *bb0268* does not encode an Hfq homolog. While one could compile a short manuscript that only documents BB0268 was incorrectly annotated, we think that would not be satisfying and leave the reader wondering about the true role of BB0268.

The second story deals with the molecular analysis of FlgV. The authors convincingly demonstrate that it is a conserved part of the flagellar regulon and that the protein interacts with the flagellar basal body. I think the *flgV* phenotype is a partial down (rather than out) for motility and the motility assays would benefit from including a fully non-motile mutant for comparison. If true, the phenotype is rather subtle, and the mechanism of promoting filament completion on a subset of flagella is unclear. I note that in terms of storytelling the authors could have chosen to study FlgV by reverse genetics based on the conserved but unknown function of the protein as

easily as choosing to study FlgV by reverse genetics based on its reported role as Hfq without altering the major outcomes of the analysis.

Several other studies have examined a fully non-motile mutant (such as a *flaB* deletion). We discuss this on pages 26 and 28 and make direct comparisons between other studies and our observations with *B. burgdorferi* Δ *flgV*.

The third story deals with the role of FlgV in the *Borrelia* infection cycle. Skillfully using what appear to be remarkably well-developed tick and mouse model systems (perhaps new (?) or perhaps from the literature), the authors show that FlgV is needed for dissemination and propagation in the mouse but not the tick. The data are beautiful and the interpretations clean and careful. These results could be novel but I'd be surprised if motility defects haven't already been shown to cause a defect in mammalian infection. This story and the detailed dissection of the infection cycle seems like it could also be a stand alone work, and would also benefit from having a direct comparison to a fully non-motile mutant. It would be useful to know whether *flgV* phenocopies a full flagellar defect or manifests as an intermediate phenotype.

Thank you for these nice comments about our infection work. Δ *flgV* *B. burgdorferi* were attenuated in their ability to infect mice and ++*flgV* *B. burgdorferi* are not infectious to mice. This is in contrast to a previous study which observed that *B. burgdorferi* lacking flagellar filaments entirely (Δ *flaB*) can survive, but at a reduced burden, in ticks, but are not able to infect mice (PMID: 23529620). This is discussed on page 28 of the manuscript. We think there are exciting future follow-up studies to further understand the role of *flgV* and the timing/regulation of flagellar filament synthesis during infection.

To be clear, I think the paper could be published as is, but I feel that the excessive length and somewhat tenuous links between stories causes nuanced points to get lost and almost draws attention to weaknesses. I note that the intro contains a seemingly extraneous paragraph on FlhF/FlhG and the discussion could be trimmed down as well, since as near as I can tell, there isn't much mechanistic clarity to explain the partial filament phenotype. If I had to guess, the FlgV mutant might have a reduced rate of type III secretion (but I imagine this would be quite challenging to test in a spirochete and I'm not suggesting it is necessary for publication).

As suggested, we have removed the paragraph about FlhF/FlhG from the introduction. The discussion was not shortened to be able to address other reviewer comments. We are excited to further characterize *flgV* phenotypes in a future study.

Minor comments:

Line numbers would aid review.

Thank you for this suggestion. We have added line numbers.

Abstract: "Flagellar motors" is a bit of a loaded term in the field and could suggest that FlgV interacts with either the stators or the C-ring to control power. But the point of the paper I believe is that FlgV controls flagellar assembly and confusion could arise since the EM data places FlgV near the rotor. Consider changing to "flagella" or "flagellar basal bodies".

We agree with the reviewer that it might be better to highlight the interaction between the flagellar basal body and FlgV. It appears that there is no interaction between the stator complexes and FlgV. We have changed "flagellar motors" to "flagellar basal bodies" in the

abstract and throughout the text.

p. 10 top. I do not think that the promoter of the flgB operon is consensus for s70. Double check. The -10 looks to be consensus but the -35 is not.

-35 regions are not well-conserved in *B. burgdorferi*, as previously documented by global transcription start identification (Adams et al., 2017; PMID: 27913725).

p.12. “flagellar filamentation” is potentially problematic as filamentation is often a word to describe cell morphology (and an elongated or filamentous cell phenotype appears later in the manuscript). Consider “flagellar filament synthesis” or “expression of a regulon that includes the flagellar filament”.

We agree with the reviewer that it is better to use “flagellar filament synthesis”. This has been corrected (page 12 and also page 22,26).

p. 26. Please clarify what is meant by “intramembrane flagellar sensor”. I think it is an important idea but not quite sure what the authors are thinking in terms of FlgV.

The sensing of a signal by a membrane protein can happen on the periplasmic face, cytoplasm or within the membrane. The hypothesis that FlgV as a sensor is inferred from conservation; the existence of conserved polar residues within the TM segments is indicative that such a sensing might happen within the membrane. The text has been changed (page 26) to clarify what we meant by intra-membrane sensor.

Reviewer #3 (Remarks to the Author):

Borrelia burgdorferi (Bb) is a spirochetal bacterium that causes Lyme disease. Motility via their endoflagella is essential for migration between tick and vertebrate hosts via dense tissues, thus there is interest in understanding the mechanisms that control this essential virulence function. The current study was initially intended to further identify and assess the RNAs bound by the previously reported putative “Hfq-like” BB0268 protein. However, the authors initial studies using multiple well-controlled immunoprecipitation methods were not able to detect any significant RNA-binding properties, suggesting a mischaracterization. Multiple in silico searches for Hfq homologs in Bb provided no significant similarities, though other categories of RNA binding proteins were present. Alpha Fold predictions for BB0268 possessed no similarities to Hfq proteins and subsequent attempts to recapitulate the findings of the previous publication identifying Bb0268 as a Hfq-like proteins were unsuccessful. A subsequent genomic survey indicated that BB0268 is situated in the flgB operon together with many other motility-related genes, and that homologs exist in many other bacterial genera, with the most closely related being flaV in *Campylobacter*. This suggested that BB0268 (FlgV) is involved in motility, and subsequent studies determined that FlgV impacts cell division and motility, that it is located in the outer membrane and is associated with the flagellar motors at the poles, that certain FlaV levels are required for assembly of flagellar filaments, and that deficiencies in FlaV has significant effects on infectivity/persistence within mouse tissues. Overall, this is a very well organized and controlled set of studies, and the manuscript is very well written to describe the significance of the experimental findings. The data provided to disprove that BB0268 is a Hfq-like protein is convincing, as is the newly proposed role in flagellar organization and motility. While there appears to be a number of significant findings, including the counterpoint to the previous BB0268 characterization, there are a few issues that should be addressed so the

reader can better understand/assess the findings.

Issues

1. Introduction. The use of passive voice in the introduction leads to some slightly bulky phrasing (e.g. page 3 line 15 “Larval ticks are hatched uninfected” reads more cleanly as “Larval ticks hatch uninfected”).

We have made the suggested change and rephrased other sentences to remove instances of passive voice.

2. Page 4, line 2. The phrase “overcome host immune responses” is a bit unclear and suggestive. Can this be rephrased to better reflect the authors intent.

This phrase was removed. We now end with the statement about dissemination, which is a broad concept that would include mechanisms of immune evasion.

3. Page 11, line 13. A reference should be provided regarding the statement that the listed motility-related genes may control flagellar count.

We have cited PMID:32039533 in which FlhF is shown to regulate the number and configuration of flagella in *Borrelia*.

4. Page 13 and description of Figure 3C. Is the decreased growth of the *flgV* mutant significantly less than the WT? Should be able to perform statistics on the multiple growth curves generated to produce the graph.

We performed an additional growth curve with data collected approximately every 12 hours and calculated the growth rate and double time of WT and $\Delta flgV$ spirochetes. This demonstrated a significant, prolonged doubling time of $\Delta flgV$ spirochetes in lag phase (8.5 h), compared to WT spirochetes in lag phase (4.6 h):

	Lag Growth Rate 1×10^5 to 3×10^6 cells/ml	Doubling Time (hours)
WT #1	0.15	4.68
WT #2	0.15	4.68
WT #3	0.15	4.59
WT Average	0.15	4.65
$\Delta flgV$ #1	0.07	9.33
$\Delta flgV$ #2	0.11	6.56
$\Delta flgV$ #3	0.07	9.59
$\Delta flgV$ Average	0.08	8.49

A student t-test was performed to compare the average double time between average WT and average $\Delta flgV$ doubling time ($p = 0.0159$).

These data have been added to Fig. S4F and are now mentioned in the text (page 13).

5. Figure 5K. The authors state that there are differences in the intensity of certain bands, which is not really apparent by eye and there is no quantification. It would be prudent to perform densitometry on the multiple gels represented by this figure and acquire hard units to allow statistics.

We now report the fold change as determined by densitometry in the figure legend and reference this in the text (page 18).

6. Page 25, line 13. The authors state that “Flagella-mediated motility is likely required to efficiently separate dividing spirochetes” but provide no reference. Adding references PMID 25690096 and 25690096 may help support this statement.

We agree with the reviewer and have added three references: PMID: 25690096, 25968649, 10995478.

7. Page 28, line 10. The authors state that “This suggests that motility is not critical for *B. burgdorferi* inside ticks”. This appears to be misleading as the mutants do maintain some motility, as evidenced by the agar diffusion assay and the dissemination in the murine host. This statement should be altered for clarity.

We have altered this statement to say: “This suggests that full motility is not critical for *B. burgdorferi* inside ticks.”

Reviewer #4 (Remarks to the Author):

Reviewer #5 (Remarks to the Author):

Bacteria flagellar assembly and flagella-mediated cell mobility are critical for microorganisms' survival and pathogenesis. In this study, Philip P. Adams and co-authors combined multiple approaches to thoroughly investigate the *B. burgdorferi* gene *bb0268/flgV* in flagellar regulation and assembly, and subsequently characterized its role in *B. burgdorferi* dissemination in mice. Based on these studies, the authors have reannotated the function of the *flgV* gene. They first showed that *bb0268/flgV* is not an RNA binding protein, not an Hfq homolog that involves in flagellar regulon regulation, as claimed by a previous study, but rather a membrane-associated protein. They then demonstrated that *flgV* is co-expressed with flagellar motor/basal body and localized the position of *flgV* by using cryo-ET in the context of different isogenic strains. They showed that *flgV* localizes to the flagellar motor, likely between the C-ring and MS rings. The authors further demonstrated that *flgV* protein, either by deletion or overproduction, doesn't change the *B. burgdorferi* motor assembly, numbers, and positions, but instead affects the assembly/length of the flagellar filament, which ultimately damages the *B. burgdorferi* mobility, cell division and dissemination in mice.

Overall, the experiments are well designed, the results are convincing, and the manuscript is well-written and easy to follow. Considering the clarification of the role of *flgV* in the regulation of the pathogenetic bacteria *B. burgdorferi* flagellar filament and the substantial amount of work, this manuscript is suitable for publishing in Nature communications, in my opinion, and would be

of great interest to researchers in the flagellar field, specifically those studying *B. burgdorferi* mobility and its mediated Lyme disease. This work would pave the way for further study to elucidate the exact mechanism of flgV in the regulation of flagellar filament length. I have only one major concern and several minor points.

My major concern is about the interpretation of the cryo-ET map. As flgV contains two transmembrane helices and has a short intracellular disordered loop region, it would be a challenge to identify its precise position from the low-resolution cryo-ET map. I have no doubt that flgV is part of flagellar motor as shown by the authors using the light microscope. I also appreciate that the authors mentioning a parallel study: "FlgV forms a flagellar motor ring that is required for optimal motility of *Helicobacter pylori*," showing a similar position of CjflgV in the motor. It seems from Figure 4C right panel that flgV mediates the interaction of the C-ring and M-ring. It would be great for the authors to check and clarify whether the density is really from flgV. Or deletion of flgV changed some part of the flagellar motor, and consequently alters the density observed by the authors. Could the authors provide some evidence of direct or indirect interaction of flgV with the flagellar MS ring protein FliF through e.g., co-IP and AlphaFold2 prediction.

We agree with the reviewer that it would be challenging to resolve individual FlgV molecules due to the limited size of the protein (two transmembrane helices and a short disordered loop) by low-resolution cryo-ET. The main reason that the FlgV densities can be resolved in our low-resolution cryo-ET map is because multiple FlgV molecules form a ring at the junction between the MS-ring and the C-ring. FlgV forms a similar ring in *H. pylori*. However, there is a clear difference between the FlgV rings from *B. burgdorferi* and *H. pylori*. The FlgV ring in *H. pylori* has large and wide densities. The result is consistent with a previous report in which FlgV was shown to interact with two additional proteins (FliF and FliA) in Epsilonproteobacteria (PMID: 24961693). Therefore, the FlgV ring densities observed by cryo-ET may correspond to all three proteins in Epsilonproteobacteria. FliF and FliA are not present in *Borrelia* sp.

To further demonstrate the observed density can be attributed to FlgV, we analyzed the basal bodies of *flgV-gfp* spirochetes. The FlgV C-terminal fusion to GFP increases the size of FlgV by 237 aa. We observed a significant increase in the FlgV-associated cryo-ET density for WT *flgV-gfp* samples (green arrows; Fig. 4G). These data support the model that FlgV localizes to the flagellar basal body between the C and MS rings of the flagellar rotor (Fig. 4C), as a ring of multiple FlgV molecules.

We have added additional statements to the discussion regarding this point (page 27).

Minor points

In Fig.2, the authors classify flgV genes into three types through analyzing their genome context. The author should also build a phylogenetic tree, at least using the sequences in figure S6A, to analyze the evolution of flgV genes from different bacteria species.

The three families are extremely remote and not really easily unified by sequence. Therefore, making a tree for a protein that is compositionally biased and often variable in length can be fraught. Nevertheless, we made a phylogenetic tree (Fig. S7) and provide a statement in text describing the phylogenetic analysis (page 12).

In Fig. 4C, the flagellar stator units need to be annotated.

We agree and the figure has been updated.

In Fig. S3, 'AlphaFold2 secondary structure predictions' need to be corrected as 'AlphaFold2 tertiary structure predictions'. AF2 predicts protein 3-D dimensional folding. The authors need to correct this in the main text on page 8, in three positions.

This has been corrected.

The author should also show in Fig. S3, the AF2 predicated dimer of flgV, and map the position of the conserved Arg highlighted in Fig. S6A, to infer if the Arg is critical for flgV dimerization, since the author has observed in the denatured gel that some flgV exists as a dimer.

Thank you for this suggestion. We have updated Fig. S3, regenerating the AlphaFold predictions using the newly published AlphaFold 3. As the reviewer suggested, we also used AlphaFold 3 to predict a FlgV dimer and highlighted the conserved arginine (added as Fig. S3D). It should be noted that the confidence for the dimer prediction is extremely low (pLDDT less than 70). The hypothesis that the arginine residue plays a role in dimerization is intriguing. We would like to test this hypothesis in a future study.

Uniform word spelling in the text needs to be followed, e.g. either Spirochetes or Spirochaetes.

Spirochaeta is used whenever we refer to the phylum. The colloquial American form spirochete(s) is now uniformly used throughout the text when not referring to the phylum.

In some position FlgV should be written as *Italic*, and BB0628 should be written as bb0628. Please check out throughout the manuscript.

We have checked the manuscript to make sure FlgV/BB0268 is used to refer to the protein while *flgV/bb0268* is used to refer to the gene, per convention.

RESPONSE TO REVIEWERS

Reviewer #1 (Remarks to the Author):

The authors have clarified all issues raised in my review. I delightedly endorse the manuscript for publication.

Reviewer #2 (Remarks to the Author):

The authors have addressed my concerns.

Reviewer #3 (Remarks to the Author):

[No comments for authors]

Reviewer #4 (Remarks to the Author):

I co-reviewed this manuscript with one of the reviewers who provided the listed reports. This is part of the *Nature Communications* initiative to facilitate training in peer review and to provide appropriate recognition for Early Career Researchers who co-review manuscripts.

Reviewer #5 (Remarks to the Author):

The authors have addressed all my major concerns. I have only one minor question remaining: The authors mentioned that "FliF and FliA are not present in *Borrelia* sp." If so, what protein forms the MS ring in *Borrelia* sp.? FliF? Once this is clarified, I support the publication of the paper.

We appreciate the reviewer pointing out this error. FliF is present in *B. burgdorferi* and is part of the MS ring. No FliA homolog has been found in *B. burgdorferi* (PMID: 9168617). We have slightly modified the text and removed the statement about FliF. It now states on page 27: "A previous study in *Campylobacter jejuni* reported FlgV interacts with additional flagellar proteins. We did not identify a FlgV homolog in flagellated-Enterobacteriaceae, all of which harbor σ^{28} . Yet, there might be other functional analogs of FlgV that were not found by our analysis. The mechanistic details of how FlgV impacts flagellar filamentation, possible FlgV interacting partners, and the differences across bacterial species are exciting directions for future work."